# Are peer reviewers influenced by their work being cited?

Adrian Barnett*

School of Public Health and Social Work, Queensland University of Technology, Brisbane, Australia

## eLife Assessment

This **important** study explored a number of issues related to citations in the peer review process. An analysis of more than 37000 peer reviews at four journals found that: (i) during the first round of review, reviewers were less likely to recommend acceptance if the article under review cited the reviewer's own articles; (ii) during the second and subsequent rounds of review, reviewers were more likely to recommend acceptance if the article cited the reviewer's own articles; (iii) during all rounds of review, reviewers who asked authors to cite the reviewer's own articles (a practice known as 'coercive citation') were less likely to recommend acceptance. However, when an author agreed to cite work by the reviewer, the reviewer was more likely to recommend acceptance of the revised article. The evidence to support these claims is **convincing**.

**Abstract** Peer reviewers sometimes comment that their own journal articles should be cited by the journal article under review. Comments concerning relevant articles can be justified, but comments can also be unrelated coercive citations. Here, we used a matched observational study design to explore how citations influence the peer review process. We used a sample of more than 37,000 peer reviews from four journals that use open peer review and make all article versions available. We find that reviewers who were cited in versions after version 1 were more likely to make a favourable recommendation (odds ratio = 1.61; adjusted 99.4% CI: 1.16–2.23), whereas being cited in the first version did not improve their recommendation (odds ratio = 0.84; adjusted 99.4% CI: 0.69–1.03). For all versions of the articles, the reviewers who commented that their own articles should be cited were less likely to recommend approval compared to the reviewers who did not, with the strongest association after the first version (odds ratio = 0.15; adjusted 99.4% CI: 0.08–0.30). Reviewers who included a citation to their own articles were much more likely to approve a revised article that cited their articles compared to a revised article that did not (odds ratio = 3.5; 95% CI: 2.0–6.1). Some reviewers' recommendations depend on whether they are cited or want to be cited. Reviewer citation requests can turn peer review into a transaction rather than an objective critique of the article.

## Introduction

In 2024, a published peer-reviewed article included this remarkable sentence: 'As strongly requested by the reviewers, here we cite some references (35-47) although they are completely irrelevant to the present work' (*Yang et al., 2024*). This was a rare public example of coerced citations, where a reviewer exploits the peer review process to increase their citation counts and hence further their own career (*Seeber et al., 2019*; *Cranford, 2020*; *Burton et al., 2024*). Reviewers should be relevant experts, so some suggestions to cite their articles will be appropriate. However, excessive citation requests or requests to cite unrelated articles are unethical (*Teixeira da Silva, 2017*; *Committee on*

*For correspondence:
a.barnett@qut.edu.au

Competing interest: The author declares that no competing interests exist.

**eLife digest** Peer review is an integral part of scientific publishing in which active researchers – who may also serve as editors at scientific journals – review and assess the standard of submitted research. Peer review is therefore central to science, as it determines which articles are published in high-profile journals, and in turn, influences the careers of scientists.

Reviewers are expected to be relevant experts in the field of research they are reviewing. But this can sometimes create a conflict of interest as the reviewers' own articles may be cited in the manuscript under review, which could influence their peer review. Conversely, reviewers whose work is not cited may feel that relevant work has been overlooked and may suggest their own papers be added – a practice that has been exploited in the past.

To find out whether citations influence peer review recommendations, Barnett analysed more than 37,000 peer reviews from four journals that operate fully open peer review, making all versions of submitted manuscripts and reviews publicly available. Using a matched design, he compared two or more reviewers who evaluated the same manuscript.

The analysis showed that being cited in the first round of review did not increase the likelihood of a favourable recommendation. However, in the second round of review, reviewers who were cited were more likely to approve the article. Contrary, reviewers who requested a citation to their own work were much less likely to approve the article.

However, a similar pattern was observed for reviewers who suggested citing work other than their own. This indicates that some citation requests may reflect legitimate concerns about missing context rather than purely self-serving behaviour.

These findings provide valuable insights into how the peer review process can be improved. Some journals – including those analysed by Barnett have already taken steps to reduce inappropriate requests for citations from reviewers. Other journals could consider implementing automated systems to flag self-citation requests, which could help improve the fairness and integrity of peer review systems, ultimately benefiting both scientists and science.

*Publication Ethics, 2019*; *Wren et al., 2019*; *Hamilton et al., 2020*; *Mehregan and Moghiman, 2024*). Coerced citations can also come from editors trying to boost their journal's ranking (*Martin, 2013*; *Heneberg, 2016*; *Fong et al., 2023*).

Coerced citations are reported as a common problem in peer review. In author surveys, two-thirds reported pressure from peer reviewers to cite unrelated articles (*Singh Chawla, 2019b*) and 23% had experienced a reviewer that 'required them to include unnecessary references to their publication(s)' (*Resnik et al., 2008*). Publishers have investigated whether 'hundreds of researchers' have manipulated the peer review process to increase their own citations (*Singh Chawla, 2019a*). Some reviewers may be exploiting their power over authors who 'have a strong incentive to […] accept all "suggestions" by the referees even if one knows that they are misleading or even incorrect' (*Frey et al., 2009*).

As reviewers are often in the same field as the article's authors, they may already be cited in the article without the need for coerced citations. Reviewers who are cited may give a more favourable peer review and be more willing to overlook flaws (*Schriger et al., 2016*; *Stelmakh et al., 2023*). Some authors may try to exploit this using 'referee baiting' (*Cranford, 2020*) or 'flattery citations' (*Frandsen and Nicolaisen, 2011*) by favourably citing a reviewer's work.

The interactions during peer review between authors and reviewers can determine whether an article is accepted (*Smith, 2006*) and what results are included in the published version (*Bohorquez et al., 2025*). Given the importance of peer review for science, studies that examine how peer review works in practice are needed (*Lee et al., 2013*; *Schmidt et al., 2018*; *Tennant and Ross-Hellauer, 2020*; *Aczel et al., 2025*; *Vendé et al., 2025*). Here, we examine interactions between peer reviewers and authors using four journals that publish all article versions and all peer reviews. We had two research questions:

1. Do peer reviewers give a more or less favourable recommendation when they are cited in the article?
2. Do peer reviewers give a more or less favourable recommendation when their review includes a citation to their own articles?

**Table 1.** Descriptive statistics for the articles and peer reviews.
Q1 = first quartile, Q3 = third quartile.

| Variable | Level/statistics | Result |
|---|---|---|
| Number of reviews | n | 37,332 |
| Year | Median [Q1, Q3] | 2022 [2019, 2024] |
| Journal, n (%) | F1000Research | 24,132 (65) |
| | Wellcome Open Research | 8697 (23) |
| | Open Research Europe | 2789 (7) |
| | Gates Open Research | 1714 (5) |
| Role, n (%) | Reviewer | 34,904 (93) |
| | Co-reviewer | 2428 (7) |
| Reviewer's recommendation, n (%) | Approved | 19,984 (54) |
| | Reservations | 14,379 (38) |
| | Not approved | 2969 (8) |
| Article version, n (%) | 1 | 26,474 (71) |
| | 2 | 8995 (24) |
| | 3+ | 1863 (5) |
| Number of papers cited in article | Median [Q1, Q3] | 24 [14, 38] |
| Any citations to reviewer, n (%) | No | 32,375 (87) |
| | Yes | 4957 (13) |
| Any papers cited by reviewer, n (%) | No | 31,546 (84) |
| | Yes | 5786 (16) |
| Any citations to the reviewer's articles | No | 35,023 (94) |
| | Yes | 2309 (6) |
| Reviewer's publication count | Median [Q1, Q3] | 55 [24, 118] |
| Reviewer's country (top five only) | USA | 7655 (21%) |
| | United Kingdom | 4137 (11%) |
| | India | 2472 (7%) |
| | Italy | 1368 (4%) |
| | Australia | 1349 (4%) |
| Number of words in the review | Median [Q1, Q3] | 202 [67, 411] |

## Results

A flow chart of the included reviews is shown in *Appendix 2—figure 1*. The final sample size was over 37,000 reviews. There were more than 3500 articles that were not included because they had not yet been peer reviewed, especially recent articles. More than 2000 reviewers did not have a record in *OpenAlex* and so could not be included. These missing reviewers were more likely to be from older articles and more likely to be co-reviewers.

Descriptive statistics on the included reviews are in *Table 1*. The reviewers were cited at least once in 13% of the articles and 6% of the reviews included a self-citation. Most reviews recommended 'Approved' (54%), with only 8% recommending 'Not approved' which is low compared with many journals; however, 40–50% of submissions are rejected before articles are sent for peer review (personal communication, *F1000* staff).

The reviewers were relatively experienced, with a median number of papers of 55.

The binary predictor for citations of 'any versus none' had a generally better fit to the data compared to the linear predictor (*Appendix 3—table 1*). This indicates that for most reviewers, receiving any citation is important, and there is no linear increase for two or more citations. The following results are for the binary predictor 'any versus none', with the results using a linear predictor in *Appendix 4— figure 1*.

Reviewers who were cited were more likely to approve the article, but only after version 1 (*Figure 1* and *Table 2*). If a reviewer was cited in any versions after version 1, the odds ratio for recommending Approved versus Reservations or Not approved was 1.61 (adjusted 99.4% CI 1.16–2.23).

Reviewers who included a citation to their own articles were much less likely to approve the article for all versions (*Figure 2* and *Table 2*). The odds ratio for recommending Approved versus Reservations or Not approved was 0.57 (99.4% CI 0.44–0.73) for version 1 and strengthened to 0.15 (99.4% CI 0.08–0.30) for versions 2+. The less favourable recommendation was only for the approval of the article and the odds ratios for Approved or Reservations versus Not approved were much closer to 1.

In an unplanned analysis, we examined the behaviour of reviewers in the first two versions of the article. We examined the 441 reviews where the reviewer was not cited in version 1 of the article and included a citation to their own articles in their first review. The reviewers who were then cited in version 2 recommended approval for 92% compared to 76% for reviewers who were not cited (odds ratio = 3.5, 95% CI: 2.0–6.1). This analysis did not use matching.

In an unplanned analysis, we examined whether the reviewers' recommendations depended on whether their review included citations to articles other than their own. Reviewers who included citations in their review were much more likely not to approve the article (*Figure 3*), which was similar to the association with citations to reviewers' own articles (*Figure 2*). However, reviewers who included citations to articles other than their own were also much more likely to recommend 'Not approved', as shown by the lower odds of 'Approved' or 'Reservations' versus 'Not approved'. This association was not seen using citations to reviewers' own articles (*Figure 2*).

## Sensitivity analyses

The odds ratios when including co-reviewers with reviewers were similar to the odds ratios when using reviewers only (*Appendix 5—figures 1 and 2*).

We found no evidence that the reviewers' publication numbers or country confounded the associations between citations and recommendations (*Appendix 6—figures 1–4*).

## Text analyses of reviewers' comments

A random sample of how reviewers included citations to their own articles found some vague justifications (*Appendix 7—table 1*); for example, 'Here are some additional publications you might consider referencing'. Other sentences adhered to the publisher's guidelines for reviewers, as specific reasoning was provided for citations to their own articles (*Wren et al., 2019*). One reviewer thanked the authors for a previous citation. Three reviews did not have a relevant sentence. One reviewer likely used AI to write their review as it included the phrase 'Certainly! Here are some potential review questions for the manuscript' (*Monadhel et al., 2023*); this review included six self-citations with no justifications.

Reviewers who included a citation to their own articles or other articles were more likely to use the words 'need' and 'please' when not approving the article (*Figure 4*). In contrast, the words 'genome' and 'well' were the most strongly associated with the reviewers' approval.

To examine how often open peer reviews were viewed, we took a random sample of 200 reviews from the four journals and found that, on average, they were viewed just 1.2 times per year (*Appendix 8—figure 1*).

## Discussion

Our results provide evidence that some reviewers have a transactional view of peer review, with their final approval dependent on citations to their work. Some reviewers may be exploiting the pressure on authors to 'publish or perish'. Under this pressure, many authors may oblige and add the suggested citations, especially since adding another citation may only require a minor edit to their article (*Oviedo-García, 2024*). Both sides gain from this transaction, as the authors get an indexed publication and the reviewer gets a citation.

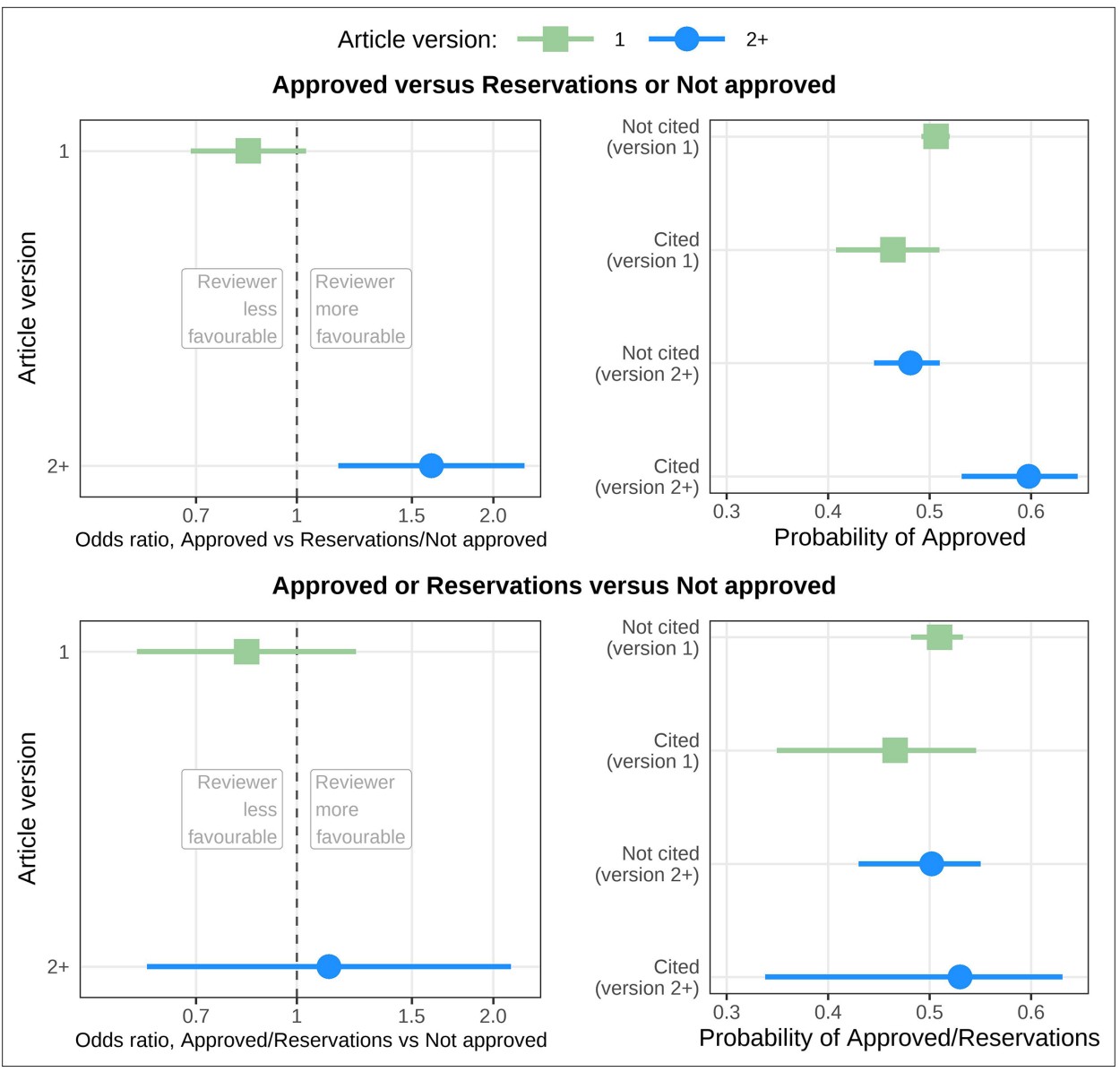

**Figure 1.** Odds ratios and probabilities for reviewers giving a more or less favourable recommendation depending on whether they were cited in the article. Top left: Odds ratios for reviewers giving a more favourable (Approved) or less favourable (Reservations or Not approved) recommendation depending on whether they were cited in the article. Reviewers cited in later versions (blue) were more likely to make a favourable recommendation (odds ratio = 1.61; adjusted 99.4% CI: 1.16–2.23), whereas being cited in the first version (green) did not improve their recommendation (odds ratio = 0.84; adjusted 99.4% CI: 0.69–1.03). Top right: Same results as top left displayed as conditional probabilities. From the top, the lines show the within-strata probability of a reviewer approving: a version 1 article in which they are not cited (0.51; adjusted 99.4% CI: 0.49–0.52); a version 1 article in which they are cited (0.46; adjusted 99.4% CI: 0.41–0.51); a version 2 (or higher) article in which they are not cited (0.48; adjusted 99.4% CI: 0.45–0.51); and a version 2 (or higher) article in which they are cited (0.60; adjusted 99.4% CI: 0.53–0.65). Bottom left: Same estimates as top left except that a more favourable recommendation is now Approved or Reservations and a less favourable is Not approved. There was no clear association for cited reviewers in version 1 (odds ratio = 0.84; adjusted 99.4% CI: 0.57–1.23) or later versions (odds ratio = 1.12; adjusted 99.4% CI: 0.59–2.13). Bottom right: Same results as bottom left displayed as conditional probabilities. From the top, the lines show the within-strata probability of a reviewer approving: a version 1 article in which they are not cited (0.51; adjusted 99.4% CI: 0.48–0.53); a version 1 article in which they are cited (0.47; adjusted 99.4% CI: 0.35–0.55); a version 2 (or higher) article in which they are not cited (0.50; adjusted 99.4% CI: 0.43–0.55); and a version 2 (or higher) article in which they are cited (0.53; adjusted 99.4% CI: 0.34–0.63). This figure is based on an analysis of 12,051 articles and 24,677 reviews for version 1 and 6090 articles and 10,196 reviews for version 2+. In all panels, a dot or square represents a mean, and a horizontal line represents an adjusted 99.4% confidence interval.

**Table 2.** Odds ratios for reviewers giving a more (OR >1) or less (OR <1) favourable recommendation depending on whether they were cited in the article (question 1) or included citations to their own articles (question 2).
All models were split by article version.

| Research question | Article version | Outcome | OR (adjusted 99.4% CI) |
|---|---|---|---|
| | Version = 1 | Approved vs Reservations/ Not approved | 0.84 (0.69, 1.03) |
| | Version = 1 | Approved/Reservations vs Not approved | 0.84 (0.57, 1.23) |
| | Versions = 2+ | Approved vs Reservations/ Not approved | 1.61 (1.16, 2.23) |
| Reviewer cited by authors | Versions = 2+ | Approved/Reservations vs Not approved | 1.12 (0.59, 2.13) |
| | Version = 1 | Approved vs Reservations/ Not approved | 0.57 (0.44, 0.73) |
| | Version = 1 | Approved/Reservations vs Not approved | 1.11 (0.77, 1.60) |
| | Versions = 2+ | Approved vs Reservations/ Not approved | 0.15 (0.08, 0.30) |
| Reviewer cited their own articles | Versions = 2+ | Approved/Reservations vs Not approved | 0.80 (0.37, 1.74) |

A key question is whether citations to a reviewer's own articles are justified as they may highlight important errors or missing context in the article. Citations to a reviewer's own articles can be justified when the authors have made a 'large scholarly oversight' (*Hamilton et al., 2020*). To investigate this, we compared the recommendations and wording of reviewers who included citations to their own articles to reviewers who included citations to other articles. The language used by the reviewers of these two groups was similar, with a higher use of 'please' and 'need' when not approving the article (*Figure 4*). However, there was a difference between groups in their recommendations, as reviewers including citations to other articles were more likely to recommend 'Not approved' (*Figure 3*) whereas this association was not observed for reviewers including citations to their own articles (*Figure 2*). This indicates that missing citations to other articles were considered more serious than missing citations to the reviewer's articles. Reviewers who cited their own articles may have been more inclined to give authors a chance to update their article and thus potentially include the 'missing' citation(s).

Examining the context of the citations to reviewers' own articles, we found vague or non-existent justifications (*Appendix 7—table 1*), showing that some reviewers ignored the journals' guidelines to state their reasoning when including citations to their own articles. However, these examples of poor justifications do not mean that all self-citations are coercive.

For both research questions, the effects were stronger for the second and later versions of the article than for the first version. Reviewers may understand that authors may be more willing to compromise on later versions when they are closer to obtaining an indexed publication. Most researchers understand that the peer review system is imperfect and that they sometimes have to make compromises to be successful (*Smith, 2006*; *Anderson et al., 2007*). Another difference to consider is that later versions will include more articles with disagreements between reviewers and more that were not 'Approved' in the first version, as articles where two or more reviewers recommended 'Approved' may not have needed a second version.

## Potential improvements to peer review

Journals could give stronger guidance to reviewers and authors on coercive citations (*Burton et al., 2024*). However, given the limited time for peer review and the many differences in guidelines between journals (*Seeber, 2020*), most authors may not read peer review instructions. Hence, guidance alone may have limited impact.

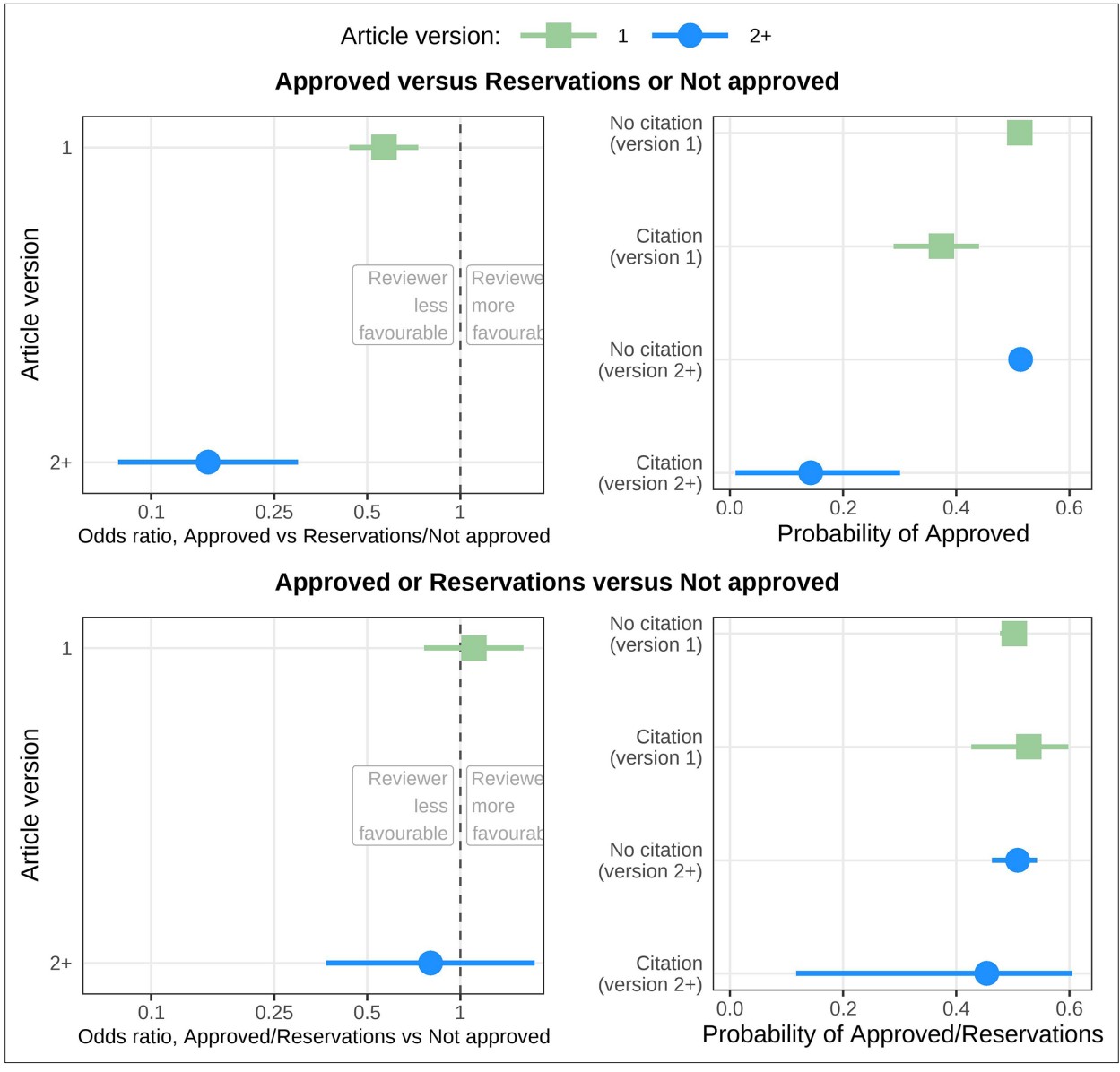

**Figure 2.** Odds ratios and probabilities for reviewers giving a more or less favourable recommendation if they included a citation to their own articles in their review. Top left: Odds ratios for reviewers giving a more favourable (Approved) or less favourable (Reservations or Not approved) recommendation depending on whether their review included a citation to their own articles. Reviewers including a citation to their own articles were less likely to make a favourable recommendation for version 1 (green; odds ratio = 0.57; adjusted 99.4% CI: 0.44–0.73) and later versions (blue; odds ratio = 0.15; adjusted 99.4% CI: 0.08–0.30). Top right: Same results as top left displayed as conditional probabilities. From the top, the lines show the within-strata probability of a reviewer approving: a version 1 article in which their review did not include a citation (0.51; adjusted 99.4% CI: 0.50–0.53); a version 1 article in which in which their review included a citation (0.37; adjusted 99.4% CI: 0.29–0.44); a version 2 (or higher) article in which their review did not include a citation (0.51; adjusted 99.4% CI: 0.49–0.53); and a version 2 (or higher) article in which in which their review included a citation (0.14; adjusted 99.4% CI: 0.01–0.30). Bottom left: Same estimates as top left except that a more favourable recommendation is now Approved or Reservations and a less favourable is Not approved. There was no clear association for reviewers who included a citation to their own articles in version 1 (odds ratio = 1.11; adjusted 99.4% CI: 0.77–1.60) or later versions (odds ratio = 0.80; adjusted 99.4% CI: 0.37–1.74). Bottom right: Same results as bottom left displayed as conditional probabilities. From the top, the lines show the within-strata probability of a reviewer approving: a version 1 article in which their review did not include a citation (0.50; adjusted 99.4% CI: 0.48–0.52); a version 1 article in which their review included a citation (0.53; adjusted 99.4% CI: 0.43–0.60); a version 2 (or higher) article in which their review did not include a citation (0.51; adjusted 99.4% CI: 0.46–0.54); and a version 2 (or higher) article in which they included a citation (0.45; adjusted 99.4% CI: 0.12–0.61). This figure is based on an analysis of 12,078 articles and 24,732 reviews for version 1 and 6101 articles and 10,213 reviews for version 2+. In all panels, a dot or square represents a mean, and a horizontal line represents an adjusted 99.4% confidence interval.

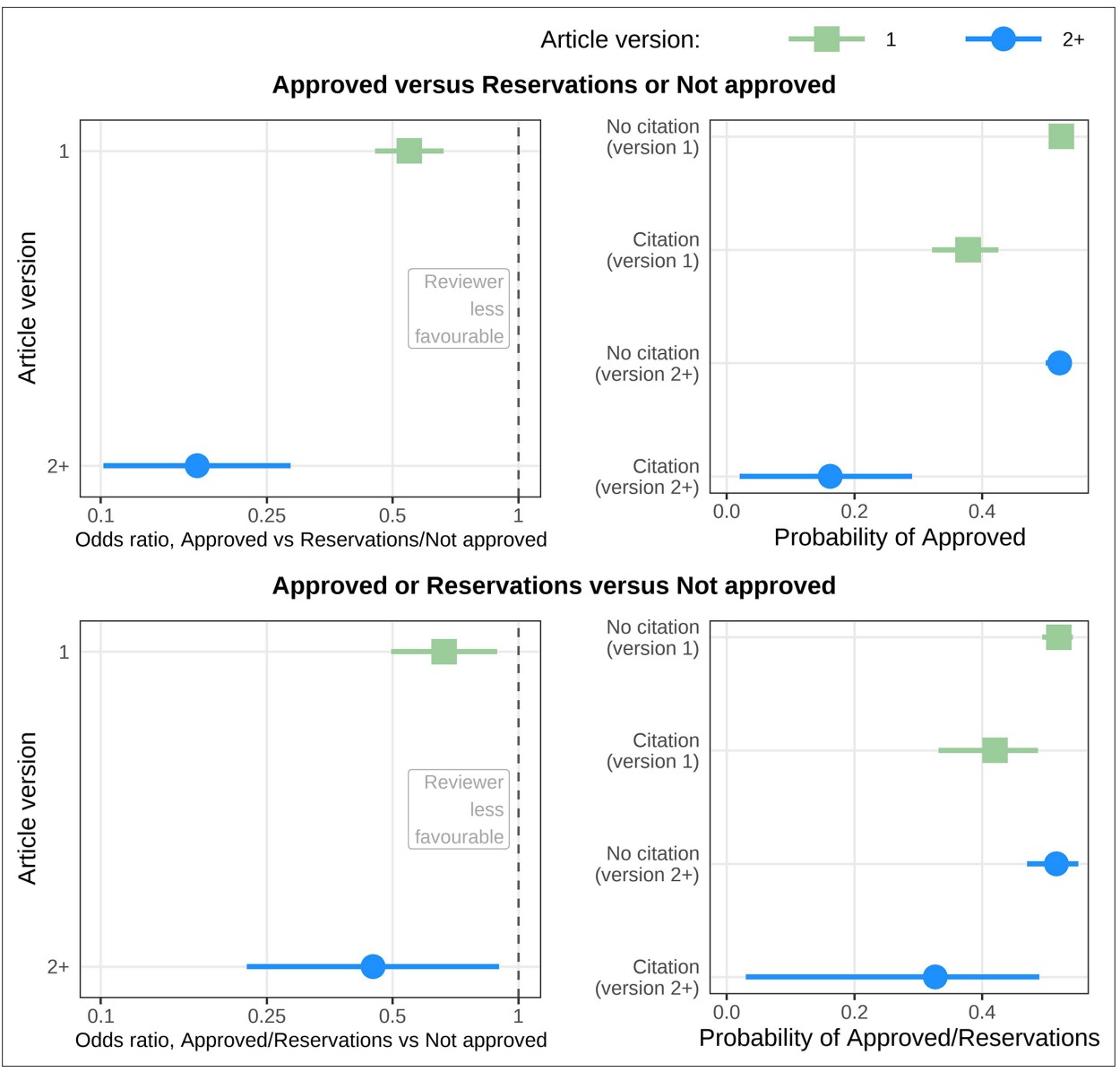

**Figure 3.** Odds ratios and probabilities for reviewers giving a more or less favourable recommendation depending on if they included citations to articles other than their own in their review. Top left: Odds ratios for reviewers giving a more favourable (Approved) or less favourable (Reservations or Not approved) recommendation depending on whether their review included a citation to articles other than their own. Reviewers including citations to other articles were less likely to make a favourable recommendation for version 1 (green; odds ratio = 0.53; adjusted 99.4% CI: 0.44–0.64) and later versions (blue; odds ratio = 0.18; adjusted 99.4% CI: 0.10–0.30). Top right: Same results as top left displayed as conditional probabilities. From the top, the lines show the within-strata probability of a reviewer approving: a version 1 article in which their review did not cite other articles (0.53; adjusted 99.4% CI: 0.51–0.54); a version 1 article in which their review cited other articles (0.37; adjusted 99.4% CI: 0.31–0.42); a version 2 (or higher) article in which their review did not cite other articles (0.52; adjusted 99.4% CI: 0.50–0.54); and a version 2 (or higher) article in which in which their review cited other articles (0.17; adjusted 99.4% CI: 0.02–0.30). Bottom left: Same estimates as top left except that a more favourable recommendation is now Approved or Reservations and a less favourable is Not approved. Reviewers including citations to other articles were less likely to make a favourable recommendation for version 1 (odds ratio = 0.62; adjusted 99.4% CI: 0.46–0.84) and later versions (odds ratio = 0.34; adjusted 99.4% CI: 0.16–0.73). Bottom right: Same results as bottom left displayed as conditional probabilities. From the top, the lines show the within-strata probability of a reviewer approving: a version 1 article in which their review did not cite other articles (0.52; adjusted 99.4% CI: 0.49–0.54); a version 1 article in which their review cited other articles (0.41; adjusted 99.4% CI: 0.31–0.48); a version 2 (or higher) article in which their review did not cite other articles (0.52; adjusted 99.4% CI: 0.47–0.55); and a version 2 (or higher) article in which their review cited other articles (0.27; adjusted 99.4% CI: 0.02–0.45). This figure is based on an analysis of 12,078 articles and 24,732 reviews for version 1 and 6101 articles and 10,213 reviews for version 2+. In all panels, a dot or square represents a mean, and a horizontal line represents an adjusted 99.4% confidence interval.

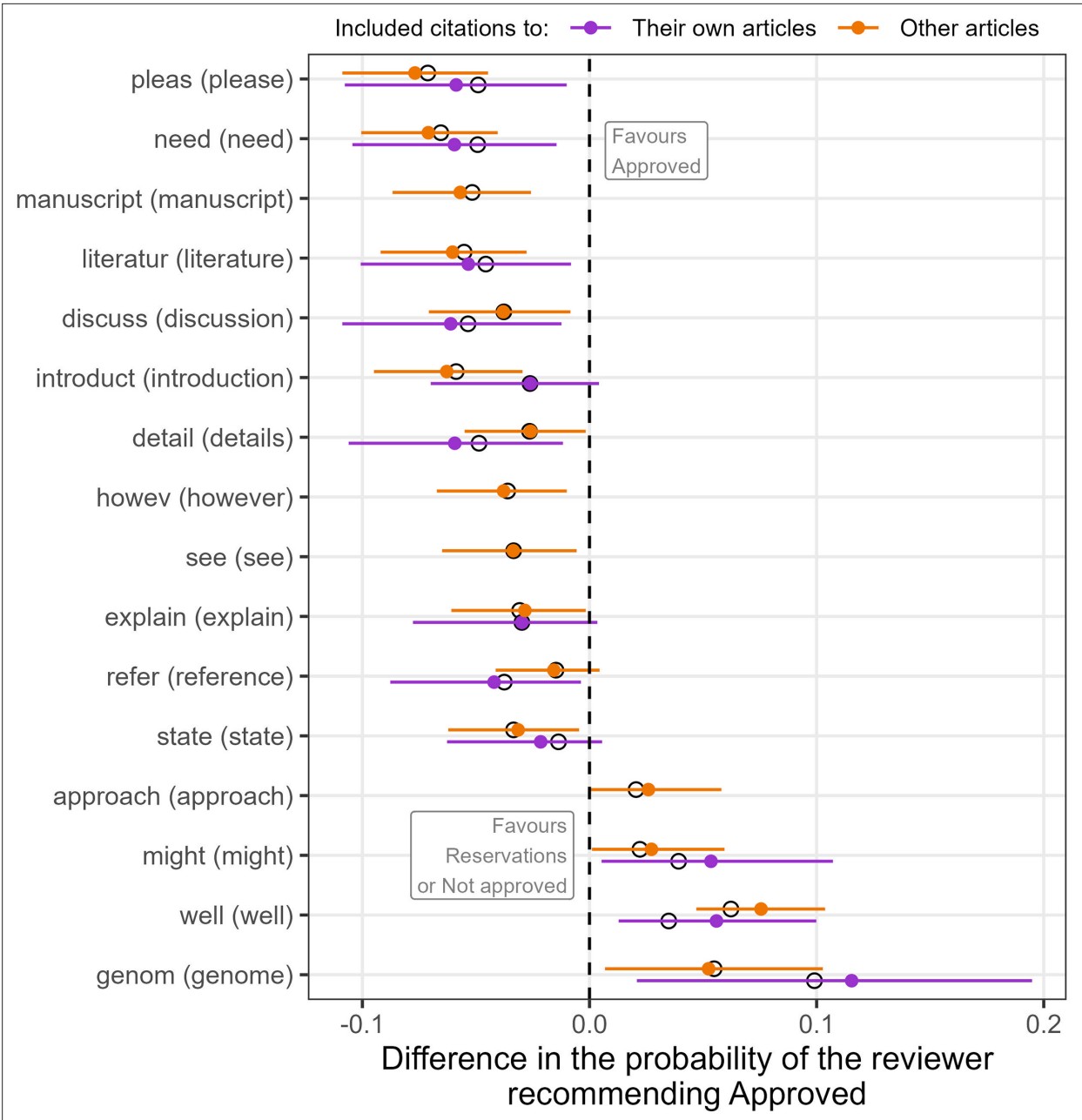

**Figure 4.** Words in the reviewers' comments that were associated with approving the article or not for reviewers who included a citation to their own articles ($n = 2025$) and reviewers who included citations to other articles ($n = 4350$). The words were selected using an elastic net that started with the 100 most commonly used review words. The estimates from the elastic net are shown as empty circles and the mean estimates and 95% credible intervals from a Bayesian model are shown as a solid circle and horizontal line. Words are shown if the probability of a non-zero mean was over 0.95 for either reviewers who cited their own articles or reviewers who cited other articles. Four words were selected by the elastic net for the reviewers who cited other articles but not by the elastic net for reviewers who cited their own articles. The axis label shows the stemmed word and most common whole word in brackets.

One suggestion is that reviewers declare to editors when they have recommended citations to their own work (***Thombs and Razykov, 2012***). A useful innovation would be for all reviews that contain citations to the reviewer's own articles to be automatically flagged to the editors who could check if the citations are justified. We are aware of one journal where this is already happening (personal communication, Benno Torgler).

*F1000* has recently introduced checks to prevent reviewers from publishing a review with three or more citations to the reviewer's own articles. If the reviewers continue to request more than three,

then the review is examined, and if the citations are deemed inappropriate and the reviewer declines to remove them, then the review is declined.

Open peer review has been suggested as a way to reduce coercive citations (*Thombs and Razykov, 2012*; *Wren et al., 2019*). However, our results from four journals that use open review show that it is not a perfect antidote, although the problem could be worse in journals using blinded peer review. The transparency of open peer review should prevent reviewers from leaving self-serving comments; however, we found some dubious justifications for self-citations and blatant use of AI (*Appendix 7—table 1*). These reviewers may have rationalised that although their words are public, they are rarely scrutinised (*Appendix 8—figure 1*); hence, it was worth the risk. The assumed additional quality assurance from open peer review (*Ross-Hellauer et al., 2017*) may often be absent.

A more radical change to peer review is that the reviewers initially see a version of the article with all references blinded and no reference list; for example, 'A strong association between increased cleaning and reduced hospital infection is well established [x]'. Reviewers are asked to give an initial recommendation and comments, and then are shown the version with the full references and asked if they need to update their recommendation or provide additional comments. However, this involves more administrative work and demands more from peer reviewers. This approach could be used for particularly consequential or controversial articles. Some journals already require authors to partially blind their articles to maintain anonymous peer review; for example, the instructions from *Taylor & Francis* include blinding the authors' names in the reference list (*Taylor and Francis, 2025*).

An argument could be made for using large language models to provide peer review that is unmoved by citation flattery. However, peer review is an inherently human task by peers, and instead, we need to improve peer review rather than abdicating this often difficult and time-consuming task to machines (*Bergstrom and Bak-Coleman, 2025*).

## Related research

Previous cross-sectional studies of self-citations in reviews found at least one self-citation in 3% at a journal that used blinded peer review (*Schriger et al., 2016*), 12% at a journal that used blinded peer review (*Thombs et al., 2015*), and 12% at a journal that used open peer review (*Peebles et al., 2020*). A related study found that 15% of reviews included a self-citation and that the self-citations were highest when the reviewer recommended 'major revisions' (*Sugimoto and Cronin, 2013*). These figures are comparable with the 6% found here and indicate that most reviews do not include self-citations.

Previous surveys estimated that 14% and 20% of authors had experienced a coercive citation request from an editor (*Fong and Wilhite, 2017*; *Wilhite and Fong, 2012*), and 7% and 23% had experienced coercive citation pressure from a reviewer (*Ho et al., 2013*; *Resnik et al., 2008*). The frequency with which researchers interact with peer review means that many will encounter coercive citations at some point in their career.

A study of conference submissions estimated that reviewers who were cited gave submissions much higher scores (*Stelmakh et al., 2023*). A study of journal peer review estimated that cited reviewers scored the article higher, but with potential confounding by the quality of the article (*Schriger et al., 2016*).

A survey of authors concluded that accepting an editor's request for citations improved the chances of being accepted (*Fong et al., 2023*). Requests in later versions were more strongly associated with acquiescence, and we found a related pattern in our analysis, with reviewers who included citations to their own articles being much less likely to recommend approval for later versions (*Figure 2*).

A study examining open peer review found that requests to cite the reviewer's articles were more likely to be included than other suggested citations, indicating that many authors wanted to please the reviewer or felt pressure to do so (*Peebles et al., 2020*).

A survey of journal editors found that only 5% objected to reviewers citing their own articles, and that this should be expected as reviewers are likely to have done related work (*Hamilton et al., 2020*).

A cross-sectional study found that reviewers' citations to their own articles were more likely to have no rationale compared to other citations, suggesting that they are more likely to be unwarranted (*Thombs et al., 2015*).

## Strengths and limitations

This is an observational study, meaning we cannot rule out unmeasured confounding and should be cautious in interpreting the results.

To our knowledge, this is the first analysis to use a matched design and analysis when examining reviewer citations, and hence strongly control for any confounding by the characteristics of the authors or articles. We compared reviewers who examined an identical article; hence, the differences we found should be due to the reviewers.

Our models include measurement error, as some citations to the reviewers' work will be missed by our data collection, and some captured citations will be inaccurate (*Pavlovic et al., 2021*). We performed random data checks that showed good accuracy (*Appendix 9—figure 1*); however, we also found valid citations that were not captured by our data extraction for conference proceedings and technical reports, which are less likely to have a DOI. This measurement error would most likely underestimate a true association, as it reduced the variance in citation counts and created a regression dilution (*Clarke et al., 1999*). Our estimates will be biased if the associations between citations and reviewers' recommendations are different for publications that do not have a DOI. Reviewers should be equally happy with any citation to their work; however, some reviewers may prefer citations to indexed articles, as these are more likely to count towards their h-indices (*Fire and Guestrin, 2019*).

We examined whether citing a reviewer altered their recommendation, but did not examine the sentiment of the citation (*Tahamtan and Bornmann, 2019*). Some citations would likely have been critical of the reviewer's articles, and we would expect these to reduce the chances of a favourable recommendation. An analysis that included the sentiment of the citation would be useful, although previous research found that most citations are neutral or positive (*Tahamtan and Bornmann, 2019*).

We did not examine the authors' responses to the reviewers, but these could include important information on why a citation was included or not in a revised version of the article. A detailed analysis examining the text used in the interactions between authors and reviewers could provide valuable information about the peer review process.

Our results may not be generalisable to journals that use blinded peer review or journals that use the traditional peer review model rather than the publish–review–curate model studied here.

A previous study found that asking reviewers to consent to an open review had no important effect on the quality of the review or the reviewers' recommendation (*van Rooyen et al., 1999*).

Another potential difference is that the journals in our sample often asked the authors to suggest peer reviewers; however, this is relatively common in other journals (*Hamilton et al., 2020*).

We found a bias in our sample, as co-reviewers and reviewers from older articles were more likely to be excluded due to not having an *OpenAlex* record (*Appendix 2—figure 1*). We therefore lost more junior reviewers who were less likely to be cited. The percentage of reviews lost was 5% (2026 of 39,113), which is hopefully small enough to avoid a large bias.

## Materials and methods

### Journal selection

We studied journals from the publisher *F1000* as their journals use open peer review with signed reviewers. *F1000* journals use a publish–review–curate model (*Currie, 2024*), meaning all versions of the article are publicly available, including versions updated after peer review. This allowed us to examine the interactions between authors and reviewers throughout the peer review process. We selected four *F1000* journals that each had over 100 articles. Some characteristics of the four journals are given in *Table 3*. Three journals were created to support funders.

**Table 3.** Brief information about the four included journals from the publisher *F1000*.

| Journal title | Year started | Field(s) of research | Articles must concern research funded by |
|---|---|---|---|
| F1000Research | 2012 | All disciplines | *No restriction* |
| Wellcome Open Research | 2016 | Medicine, Genomics | Wellcome |
| Gates Open Research | 2017 | Medicine | The Gates Foundation |
| Open Research Europe | 2021 | All disciplines | European Commission |

The peer review process used by *F1000* journals differs from most standard journals. The journals do not use academic editors, but do have in-house editors who manage articles but do not make editorial decisions. This means that most interactions during peer review are between authors and reviewers directly. In-house editors perform checks prior to the first version of the article being published and at *F1000Research* this results in 40–50% submissions being rejected (personal communication, *F1000* staff).

Up to mid-2024, authors were asked to identify potential reviewers who were qualified experts with no competing interests (*F1000Research, 2025a*). Since mid-2024, reviewer identification is made in-house, although authors can suggest reviewers.

Reviewers are asked to recommend one of three categories: Approved, Approved with reservations, and Not approved. For brevity, we refer to 'Approved with Reservations' as 'Reservations'. An article is indexed once it receives two 'Approved' or two 'Reservations' and one 'Approved'. The guidelines for recommending Approved are: 'the aims and research methods are adequate; results are presented accurately, and the conclusions are justified and supported by the presented data' (*F1000Research, 2025b*). Peer reviewers are asked to assess the validity of an article's content, rather than novelty or interest levels, an approach designed to combat publication bias (*Begg and Berlin, 1988*).

All four journals have a peer reviewer code of conduct and state that reviewers should familiarise themselves with the ethical guidelines for peer reviewers by the *Committee on Publication, 2017*. The journals' guidelines for reviewers include the following: 'reviewers should explicitly state their reasoning when asking authors to cite their own work'.

## Data extraction

We extracted data on authors and articles from the *OpenAlex* database (https://openalex.org/) and directly from the four journals. *OpenAlex* combines scholarly data from multiple sources, including *ORCID* – a unique identifier for researchers, *Microsoft Academic*, *Crossref* and *PubMed*. A recent study compared *OpenAlex* with the two commonly used proprietary bibliometric databases of *Web of Science* and *Scopus* for the years 2015–2022 (*Culbert et al., 2025*). The results were mixed, but *OpenAlex* had better *ORCID* coverage and covered more Digital Object Identifiers (DOIs) – the unique identifier for publications. We accessed *OpenAlex* using the *openalexR* package (*Priem et al., 2022*; *Aria et al., 2024*). We used each journal's application programming interface (API) to extract data on the articles and peer reviews. The data were extracted in four stages:

1. Searches were made using the APIs of the four journals to find all articles published between 1 Jan 2012 and 28 May 2025, with the start date to capture all potential articles.
2. For each article, the following data were downloaded in XML format:
   - The article's publication date and version number
   - The reviewers' names and *ORCIDs* (if available)
   - The text of all reviews and the reviewers' recommendations
   - The DOIs and PMIDs (*PubMed* IDs) from the article's reference list
   - The DOIs and PMIDs of any articles cited by the reviewers. The online peer review system at *F1000* journals includes the DOI of any article cited in the review, which facilitates the identification of citations to the reviewers' articles.
3. Articles were excluded if:
   - They were not peer reviewed or had yet to receive any reviews
   - The reference list was empty
4. The reviewers' publication histories were collected from *OpenAlex* using their name, institution and *ORCID* (if available). Reviews were excluded if there was no record for the reviewer in *OpenAlex*, or if the reviewer had no published articles as there was no potential for them to be cited or request a citation to their own articles.

## Study design

We used two predictor variables about the reviewer:

- The number of times they were cited in the article (0, 1, 2, …).
- The number of times they included citations to their own articles in their review (0, 1, 2, …).

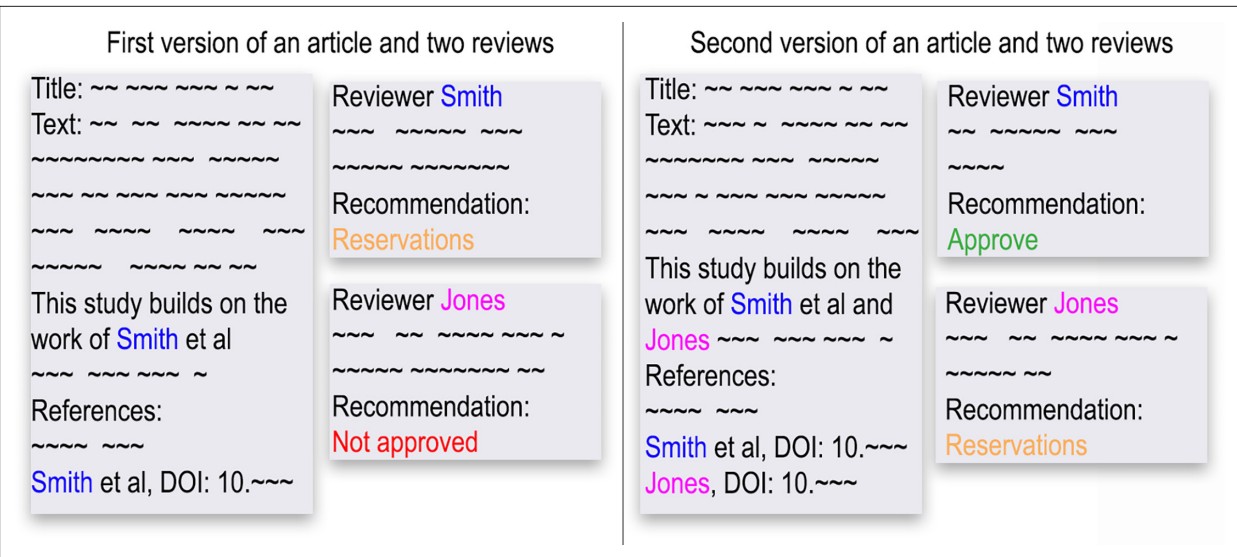

**Figure 5.** Graphical summary of the study design for research question 1 showing a dummy article and two reviews. In the first version of the article, the reviewer Smith (blue) is cited whilst Jones (purple) is not. For the second version of the article, the authors are now aware that Jones is a reviewer and Jones has been cited. The reviewers' recommendations are the outcome and are colour-coded as Not approved (red), Reservations (orange) and Approved (green). We tested whether citations to the reviewer in the article influenced their recommendation. The matched design means that only reviewers of the same article are compared (here, Smith and Jones) and the overall effect is estimated by aggregating over multiple matched comparisons. Research question 2 used the same design but examined citations to the reviewers' articles in their reviews.

We fitted both predictors as linear, but reviewers may behave differently with any citation rather than a linear change, and hence we also fitted both predictors as a binary 'none versus any' (0 versus 1, 2, …). We compared the linear and binary alternatives using the Akaike Information Criterion (AIC) to find the parameterisation that best fitted the data (*Burnham and Anderson, 2002*).

We matched on article and version to control for confounding by any characteristics of the article (*Bland and Altman, 1994*); for example, the article's topic or writing style. Hence, we compared two or more independent reviewers who considered the same article.

All analyses were stratified by article version, using the first version only or the second and subsequent versions. This is because the reviewers are unknown to the authors for the first version, but from the second version onwards, the authors will know the reviewers as the journals use signed peer reviews. This knowledge could alter the behaviour of authors and reviewers.

The study design is summarised in *Figure 5*.

## Statistical methods

We used conditional logistic regression to examine the associations between the citations to the reviewer and their ordinal recommendation (Approved → Reservations → Not approved) while matching the article and the version (*Hosmer et al., 2013*). Conditional logistic regression requires a binary dependent variable; hence, we fitted two related models that examined the odds of:

1. 'Approved' compared with 'Reservations' or 'Not approved'.
2. 'Approved' or 'Reservations' compared with 'Not approved'.

These two models tested the same hypothesis; hence, we adjusted for multiple testing. We also used repeated testing due to the stratification by article version and the two formulations of the predictors (linear or none versus any). Since we used 8 (2 × 2 × 2) tests, we displayed all the results using 99.4% confidence intervals instead of 95.0% intervals, which is a 5% type I error divided by eight tests.

In an unplanned analysis, we examined the association between the reviewer's recommendation and whether they included citations to work other than their own articles. This was added to examine differences between reviewers' citations to their own articles and other articles.

Outliers were not excluded. No data were missing in the analysis data set.

The sample size calculation is in Appendix 1.

## Text analysis

We examined how reviewers' citations to their own articles or other articles were justified and whether their wording differed according to their recommendation. For an initial view of citations to their own articles, we randomly selected 20 reviews and extracted the most relevant sentence concerning the citation.

To analyse the review text, we first extracted the 100 most commonly used words in all reviews. To standardise the text, all words were transformed into tokens, with stop-words removed and then stemmed. We then tested which of the 100 words were associated with recommending Approved versus Reservations or Not approved amongst those reviewers who included a citation to their own articles and those who included a citation to other articles. We chose the set of words using an elastic net with 10-fold cross-validation and selected a parsimonious model by using the lambda within one standard error of the minimum cross-validated error (*Zou and Hastie, 2005*; *Tay et al., 2023*). To get uncertainty intervals for the estimates, we fitted a Bayesian model with the set of words selected by the elastic net and using a sceptical Normal prior centred on zero to create shrinkage.

## Reproducibility

Research question 1 was pre-registered using *As Predicted* on 20 May 2024 (*Barnett, 2024*). Research question 2 was formulated during data collection but before any data analysis and used the same study design and statistical methods as question 1.

All data extraction and analyses were conducted using *R* version 4.4.1 (*R Development Core Team, 2024*). The data and *R* code are available on *GitHub* (*Barnett, 2025a*).

## Acknowledgements

Thanks to all four journals for making all their data openly available and easily accessible. Thanks to Robin Blythe, staff from F1000 and Paper-Wizard https://paper-wizard.com/ for providing helpful feedback on a draft of this paper.

## Additional information

### Funding
No external funding was received for this work.

### Author contributions
Adrian Barnett, Conceptualization, Data curation, Software, Formal analysis, Validation, Visualization, Methodology, Writing – original draft

### Author ORCIDs
Adrian Barnett ⓘ https://orcid.org/0000-0001-6339-0374

### Ethics
All data used are openly available and published with the expectation of post-publication scrutiny.

Joint Public Review: https://doi.org/10.7554/eLife.108748.4.sa1
Author response https://doi.org/10.7554/eLife.108748.4.sa2

## Additional files

### Supplementary files
MDAR checklist

## Data availability

All raw and analysis data used in this article are openly available here: https://github.com/agbarnett/cited_reviewers, copy archived at *Barnett, 2025b*.

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

## Appendix 1

### Sample size

We aimed for a sample size of approximately 5000 articles and assumed that half would be the first version, giving a sample size of 2500 articles for the analysis using the first version only (*Barnett, 2024*). In 1000 simulations, this gave an 89.1% power to detect an odds ratio of 1.5 using conditional logistic regression for a reviewer who recommended a higher category (Approved → Reservations → Not approved) when they were cited. We assumed that 15% of articles would include a citation to the reviewer. Eighty per cent of the simulated articles had two reviews, and the remaining 20% had three reviews. Based on preliminary data from two journals, we assumed that the reviewers' recommendations would have a ratio for Approved:Reservations:Not approved of 70:24:6.

## Appendix 2

### Included and excluded reviews

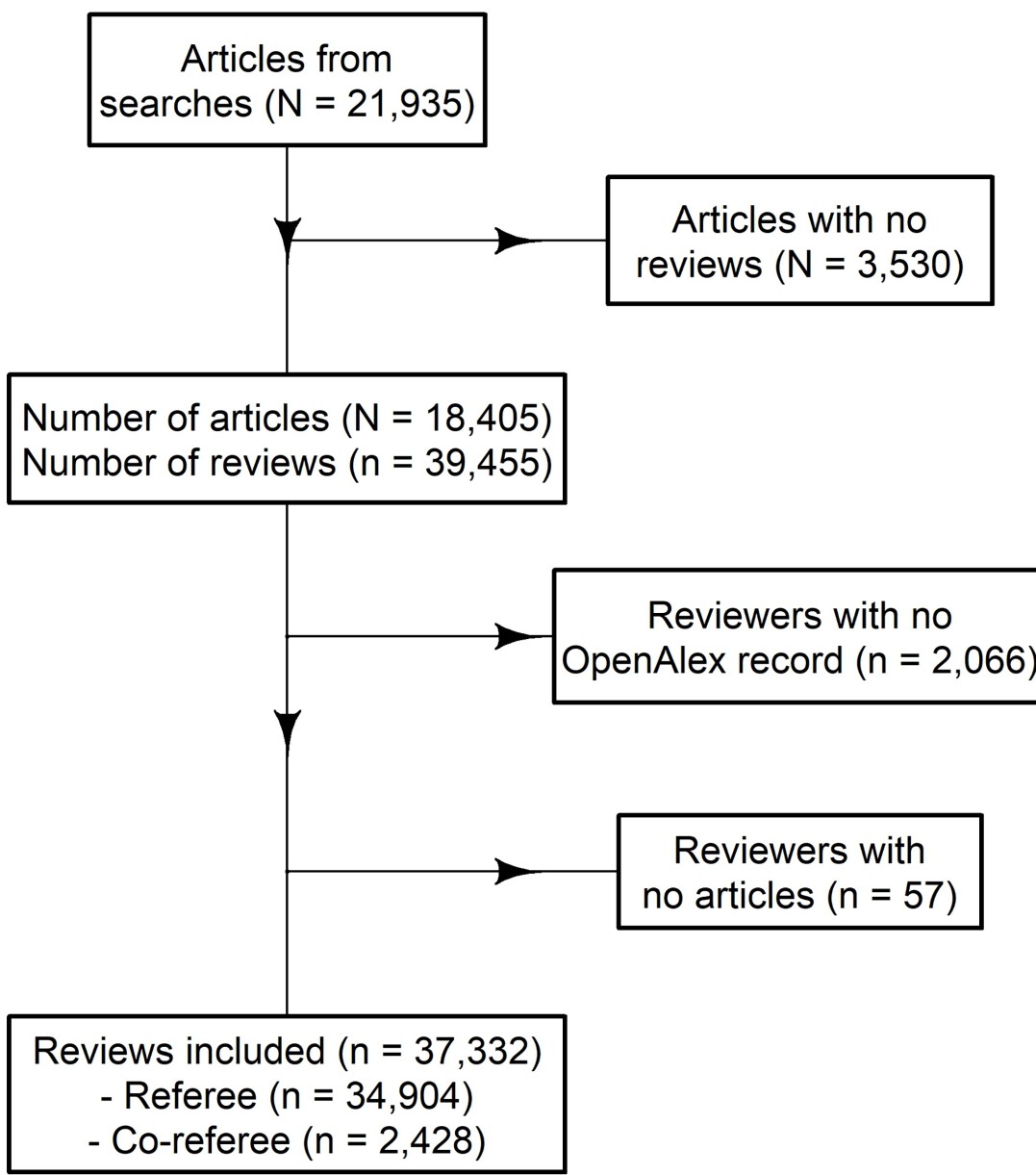

**Appendix 2—figure 1.** Flow chart of included reviews. 'N' is the number of articles and 'n' is the number of reviews.

The flow chart shows the loss of articles and reviews during the data collection process. More than 3500 articles did not have reviewers as they had yet to be peer reviewed or were Faculty Reviews that were commissioned and used a different peer review model.

More than 2000 reviewers did not have an *OpenAlex* record and therefore were excluded from the analyses. We examined the potential bias in the lost reviews by comparing their characteristics with those of the retained reviews. We used a multiple regression model with reviewer lost (yes/no) as the binary dependent variable and predictors of article version, article date, referee or co-referee, and reviewer's country. We expected many of these predictors to have little effect; therefore, we used an elastic net to reduce the number of predictors (*Zou and Hastie, 2005*). We used the 'glmnet'

package in *Tay et al., 2023*. For the binary dependent variable, 39,455 reviews were retained and 2082 (5%) were lost.

The elastic net retained two predictors. The date of the article had an odds ratio of 1.09 per year increase, which means that more recent articles were more likely to be retained, likely because the reviewer's information was more current. Referees were more likely to be retained compared to co-referees with an odds ratio of 1.79, likely because co-referees were often relatively junior and some may not have any publications.

## Appendix 3

## Model fit

**Appendix 3—table 1.** Comparing the two alternatives for the citation predictor variables using either a linear variable or a binary 'any versus none' variable.

A vs R/N = Approved vs Reservations/Not approved, A/R vs N = Approved/Reservations vs Not approved.

| Co-reviewers included | Version | Outcome | AIC | | |
|---|---|---|---|---|---|
| | | | Linear | Binary | Difference |
| No | 1 | A vs R/N | 5940.9 | 5937.4 | 3.6 |
| No | 1 | A/R vs N | 1930.3 | 1929.8 | 0.5 |
| No | 2+ | A vs R/N | 1952.4 | 1941.9 | 10.5 |
| No | 2+ | A/R vs N | 572.1 | 571.9 | 0.2 |
| Yes | 1 | A vs R/N | 5978.4 | 5975.4 | 3.0 |
| Yes | 1 | A/R vs N | 1941.1 | 1940.8 | 0.3 |
| Yes | 2+ | A vs R/N | 1963.1 | 1951.4 | 11.7 |
| Yes | 2+ | A/R vs N | 572.6 | 572.8 | –0.2 |
| No | 1 | A vs R/N | 5932.3 | 5911.1 | 21.2 |
| No | 1 | A/R vs N | 1934.1 | 1935.6 | –1.5 |
| No | 2+ | A vs R/N | 1881.4 | 1876.0 | 5.4 |
| No | 2+ | A/R vs N | 573.4 | 572.8 | 0.6 |
| Yes | 1 | A vs R/N | 5967.9 | 5944.9 | 23.0 |
| Yes | 1 | A/R vs N | 1945.4 | 1946.6 | –1.2 |
| Yes | 2+ | A vs R/N | 1917.3 | 1904.1 | 13.2 |
| Yes | 2+ | A/R vs N | 573.1 | 573.5 | –0.5 |

The AIC (Akaike Information Criterion) is a trade-off of model fit and complexity. The smaller the AIC, the better the fit. Differences of 10 are considered large (*Burnham and Anderson, 2002*).

In most cases, the difference between the linear and binary variables was small (under 5). There were four comparisons out of 16 in which the linear variable had a smaller AIC than the binary variable and all differences were small (under 2). There were four comparisons where the AIC for the binary variable was over 10 units smaller than the linear variable, indicating a large difference in model fit. In summary, using a binary predictor variable is a generally better fit to the data than using a linear variable.

## Appendix 4

### Results from using a linear predictor

The figure shows the estimates for the two research questions using a linear dose–response for citation counts instead of the binary predictor of any citation versus none. The strongest effect was a greatly reduced odds of 'Approved' for increasing citations to the reviewer's own articles. However, these estimates should be viewed with caution, as the binary predictor generally better fits the data (Supplement Model fit).

**Appendix 4—figure 1.** Estimated odds ratios for using linear citations as the predictor. The reference point is zero citations.

## Appendix 5

### Including co-reviewers

Some reviews were performed by reviewers together with co-reviewers, who were usually less experienced. Our primary analysis excluded co-reviewers, but we included them in a sensitivity analysis where we created combined versions of the two independent variables using the sum of citations to reviewers and co-reviewers, and the sum of citations to their own articles from the reviewers and co-reviewers. The results examining whether the reviewers gave a more favourable recommendation when cited (research question 1) were very similar (*Appendix 5—figure 1*).

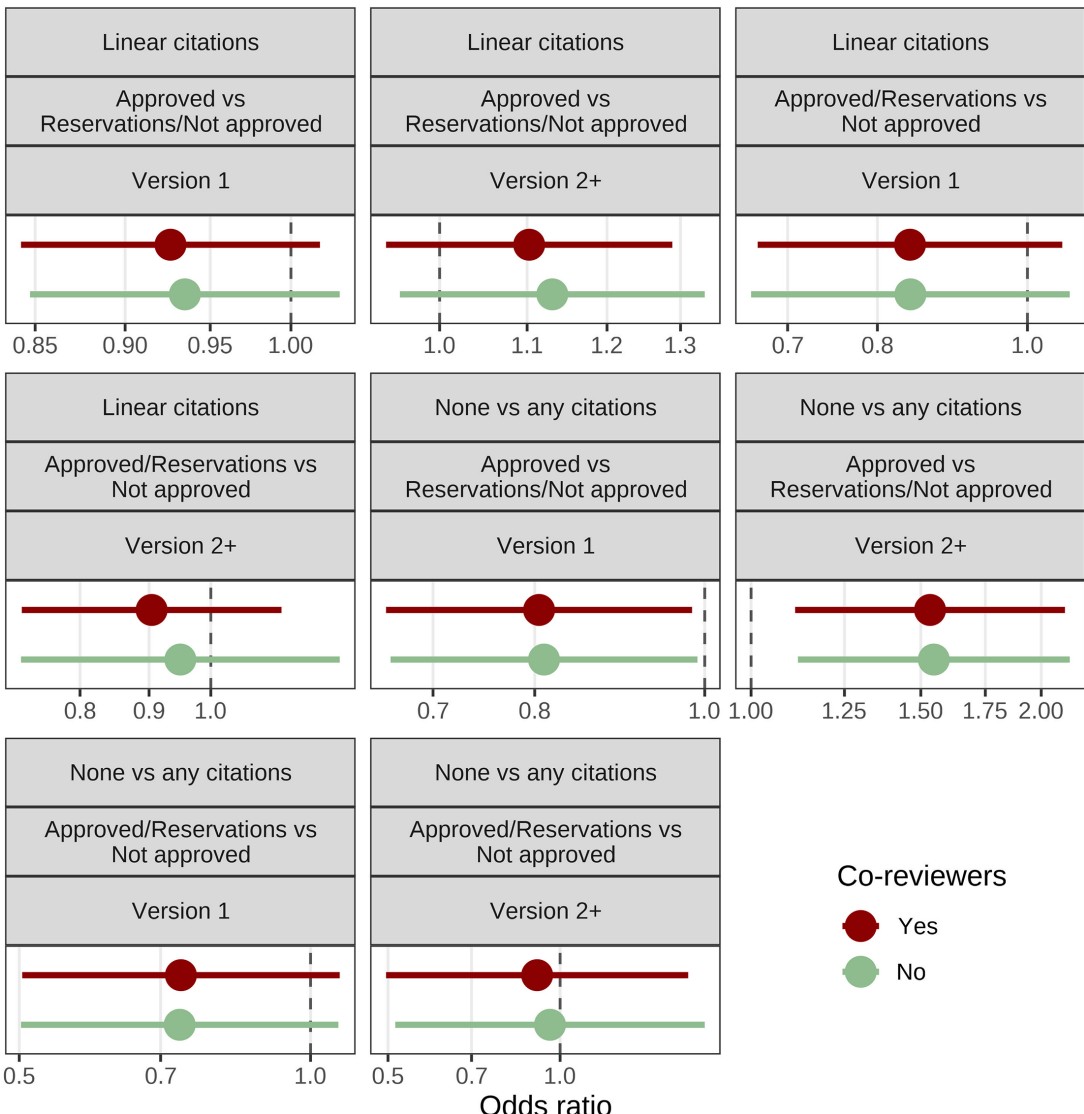

**Appendix 5—figure 1.** Results with or without co-reviewers for research question 1. Odds ratios and adjusted 99.4% confidence intervals for whether the reviewer gave a more or less favourable recommendation if they were cited. The results are shown for the combinations of predictor variables (linear or any vs none), outcome (Approved → Reservations → Not approved) and article version. The plot is designed to directly compare paired odds ratios with or without co-reviewers.

The results examining whether the reviewers gave a more favourable recommendation when they included citations to their own articles (research question 2) were mostly very similar (*Appendix 5—figure 2*). Two noticeable differences were two odds ratios where including co-reviewers somewhat reduced the strength of the association. This was for article versions 2+ and examining Approved vs Reservations or Not approved. Despite the noticeable change in the odds ratio, the interpretation

remains similar in that there was a strong reduction in the odds of a favourable recommendation when the reviewers included citations to their own articles.

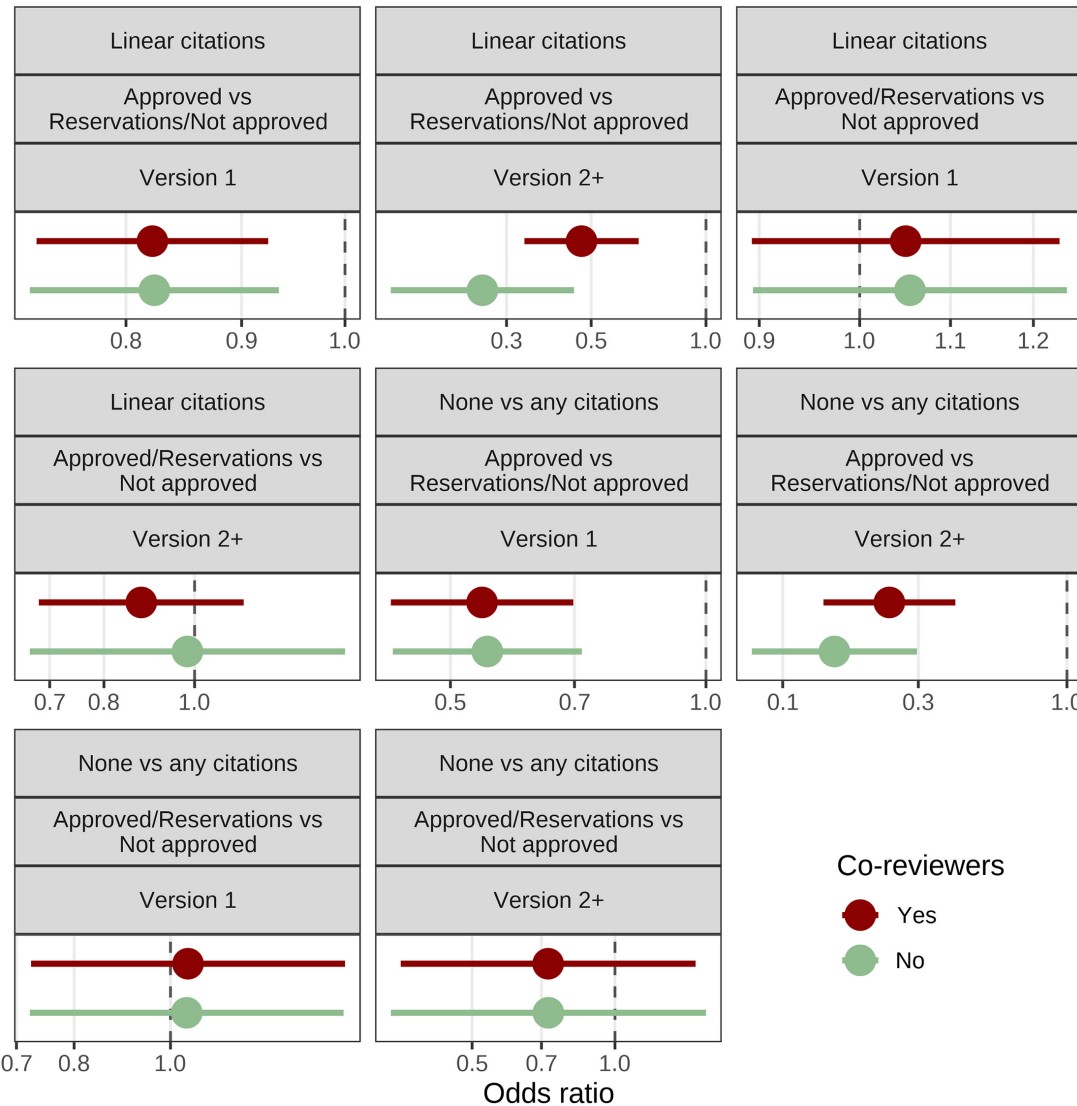

**Appendix 5—figure 2.** Results with or without co-reviewers for research question 2. Odds ratios and adjusted 99.4% confidence intervals for whether the reviewer gave a more or less favourable recommendation when they included a citation to their own articles. The results are shown for the combinations of predictor variables (linear or any vs none), outcome (Approved → Reservations → Not approved) and article version. The plot is designed to directly compare paired odds ratios with or without co-reviewers.

## Appendix 6

### Potential confounding by the reviewers' characteristics

Any confounding by the characteristics of the articles was controlled by the matched design, but confounding by the characteristics of the reviewers remains possible (*Stelmakh et al., 2023*). We considered the potential confounders of the reviewer's experience and reviewer's country. More experienced reviewers will likely be cited more often (on average) and could be more or less strict in their recommendations. The reviewer's country is a potential confounder due to large differences in citation counts by country (*Gomez et al., 2022*) and potential differences in recommendations by country (*Campos-Arceiz et al., 2015*).

### Reviewers' experience

We used the reviewer's number of published articles as a proxy for their experience. This association could be non-linear; for example, a diminishing effect for more experienced reviewers, so we examined six fractional polynomials of the reviewers' number of articles and used the AIC to select the best fit (*Royston et al., 1999*). For most models, the best fit was achieved using a log-transformation.

There was little evidence of any confounding by the reviewers' publication counts as the odds ratios were similar for both research questions (*Appendix 6—figure 1*; *Appendix 6—figure 2*). A fractional polynomial of −2 tended to show the largest difference compared to the odds ratios with no confounders; however, this transformation was not the best fit and the differences were relatively small.

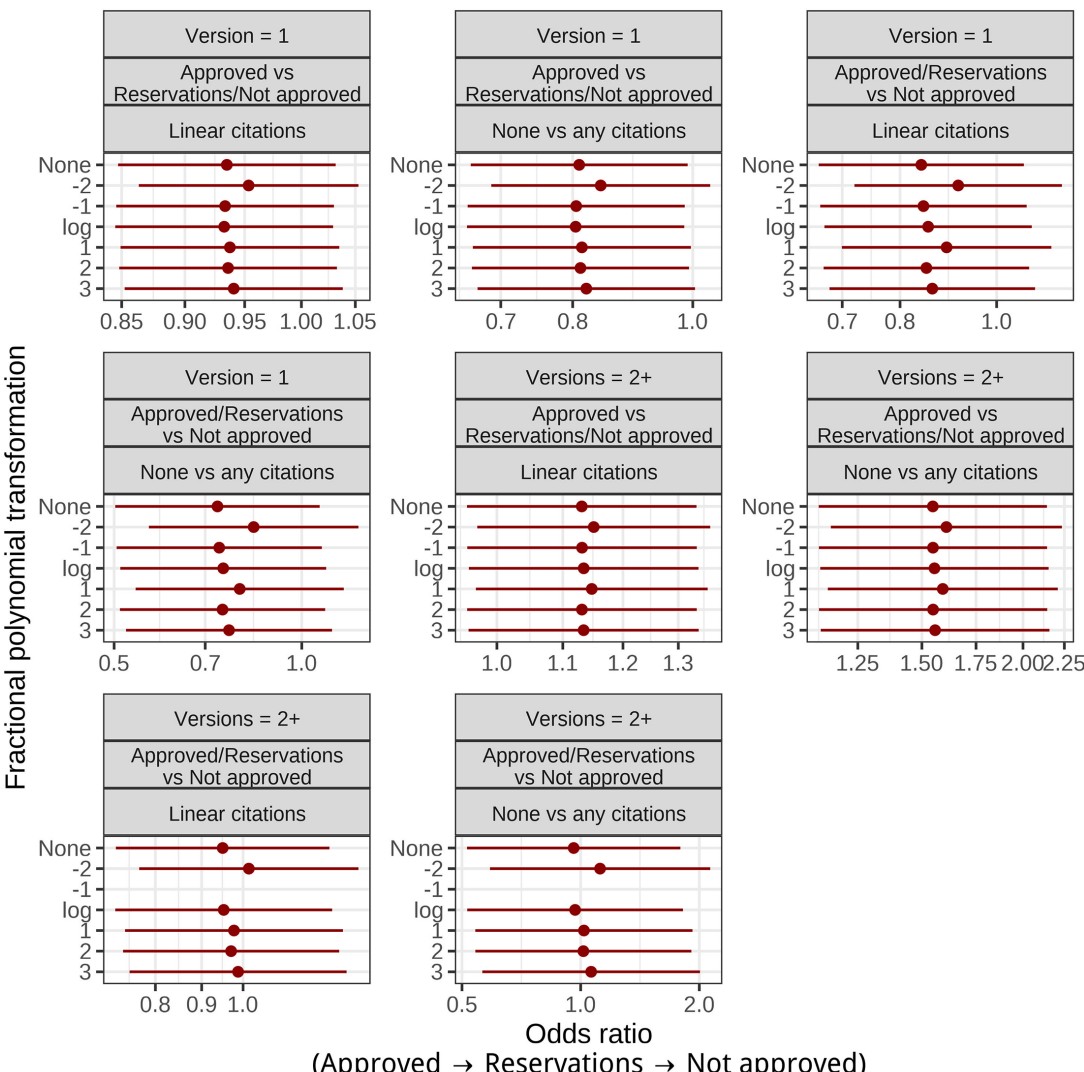

**Appendix 6—figure 1.** Examining potential confounding by reviewers' publication counts for research question 1. Odds ratios and adjusted 99.4% confidence intervals for whether the reviewer gave a more or less favourable recommendation when they were cited. We used fractional polynomials to examine a potentially non-linear association between reviewers' publication counts and recommendation. The results for 'None' are the results without the potential confounder. The results are shown for the combinations of predictor variables (linear or any vs none), outcome (Approved → Reservations → Not approved) and article version. Results are missing when the model did not converge.

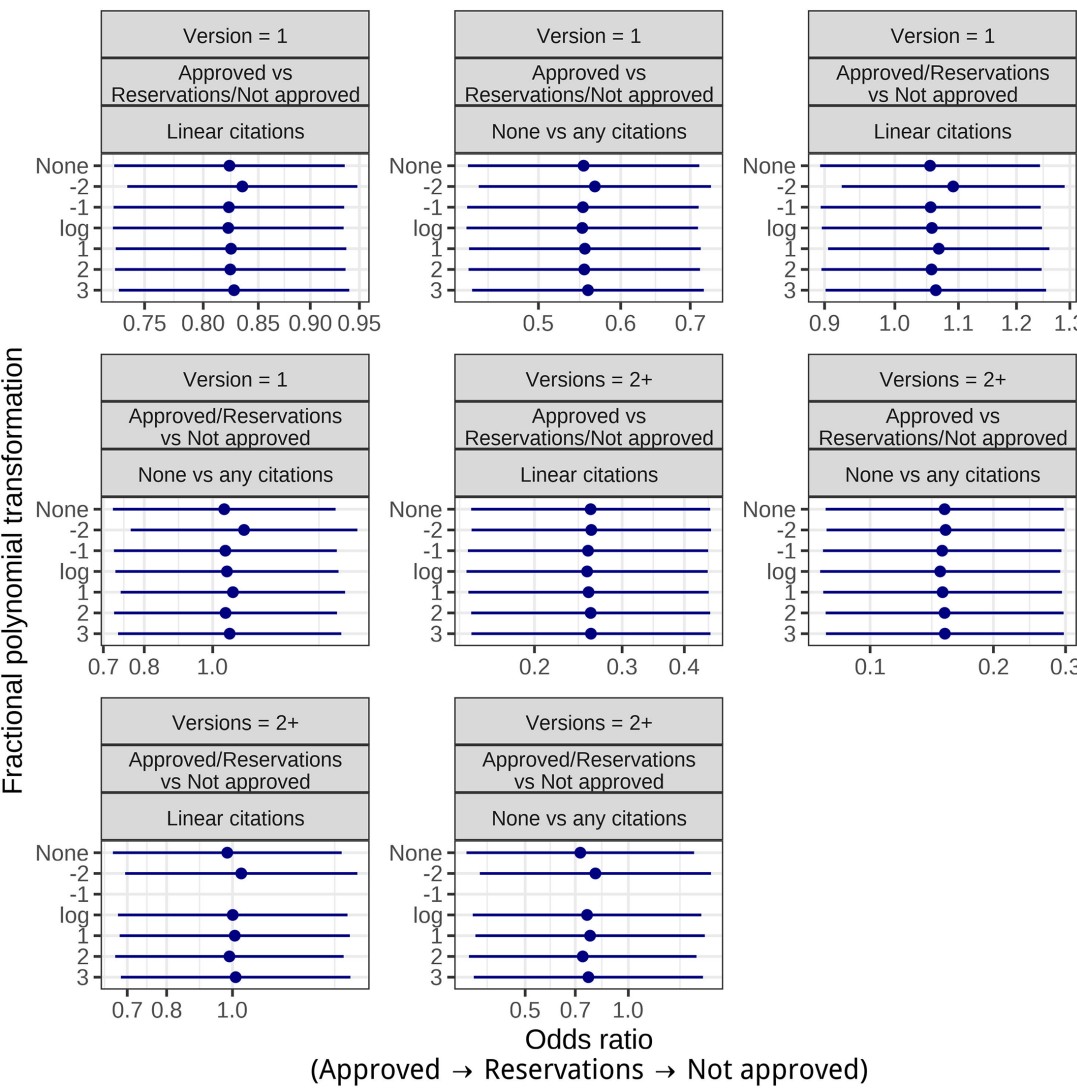

**Appendix 6—figure 2.** Examining potential confounding by reviewers' publication counts for research question 2. Odds ratios and adjusted 99.4% confidence intervals for whether the reviewer gave a more or less favourable recommendation when they included a citation to their own articles. We used fractional polynomials to examine a potentially non-linear association between reviewers' publication counts and recommendation. The results for 'None' are the results without the potential confounder. The results are shown for the combinations of predictor variables (linear or any vs none), outcome (Approved → Reservations → Not approved) and article version. Results are missing when the model did not converge.

## Reviewers' countries

We planned to use a frailty model to test for confounding by the reviewers' countries (***Therneau and Grambsch, 2013***). However, this model often failed to converge, potentially because there were many countries and some countries had relatively small numbers of reviewers. Hence, we instead used a leave-one-out analysis for each of the top 10 most common countries and determined if the results were noticeably different.

The results were generally similar regardless of which country was left out. Leaving out the USA, which was the largest country, had a relatively large effect on the odds of recommending Approved or Reservations vs Not approved for versions 2+ when using the "none vs any citations" predictor (***Appendix 6—figure 3***) and on the odds of recommending Approved or Reservations vs Not approved for versions 2+when using the "none vs any citations" predictor (***Appendix 6—figure 4***). However, neither change was substantively different from the results including all countries.

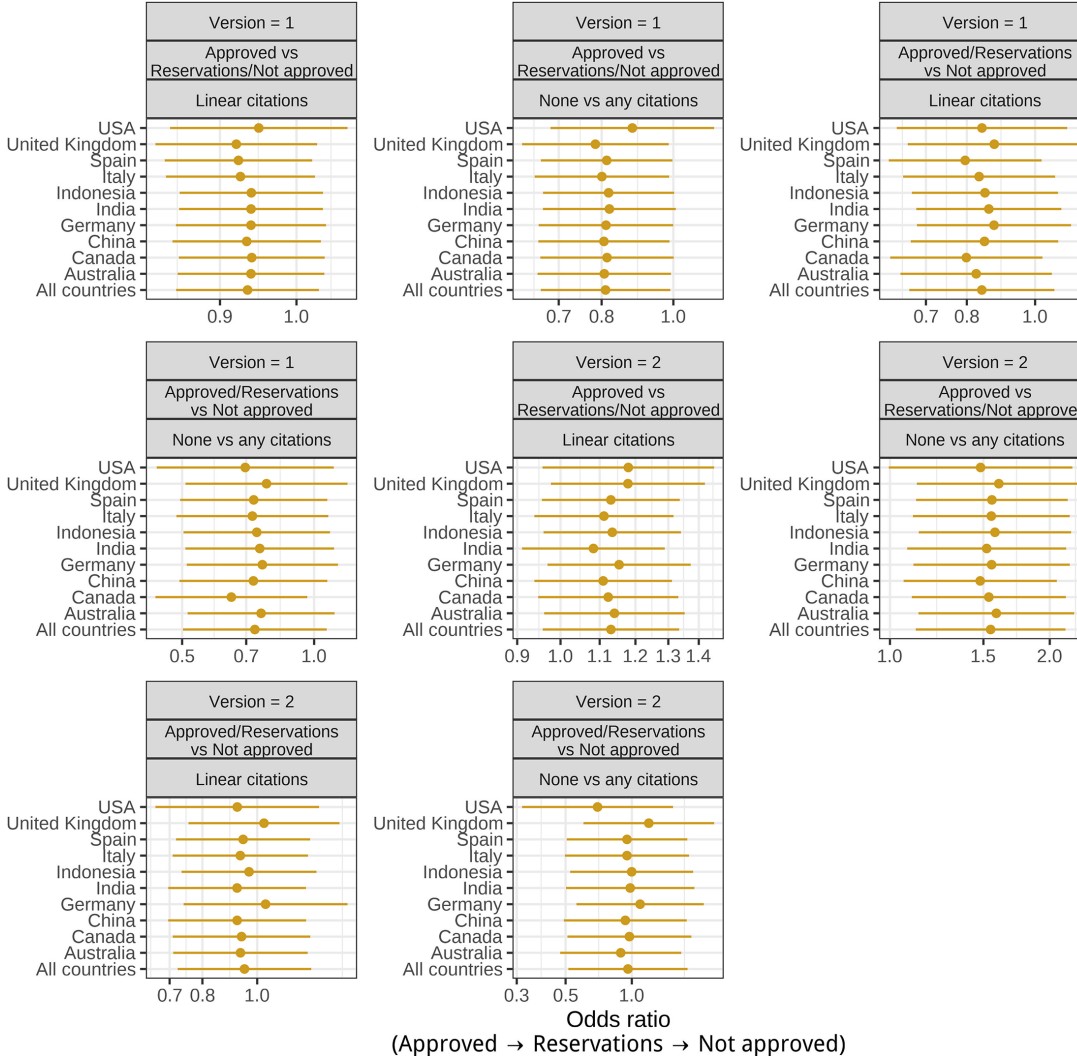

**Appendix 6—figure 3.** Leave-one-country-out sensitivity analyses for research question 1. Odds ratios and adjusted 99.4% confidence intervals for whether the reviewer gave a more or less favourable recommendation when they were cited. The results are shown for the combinations of predictor variables (linear or any vs none), outcome (Approved → Reservations → Not approved) and article version.

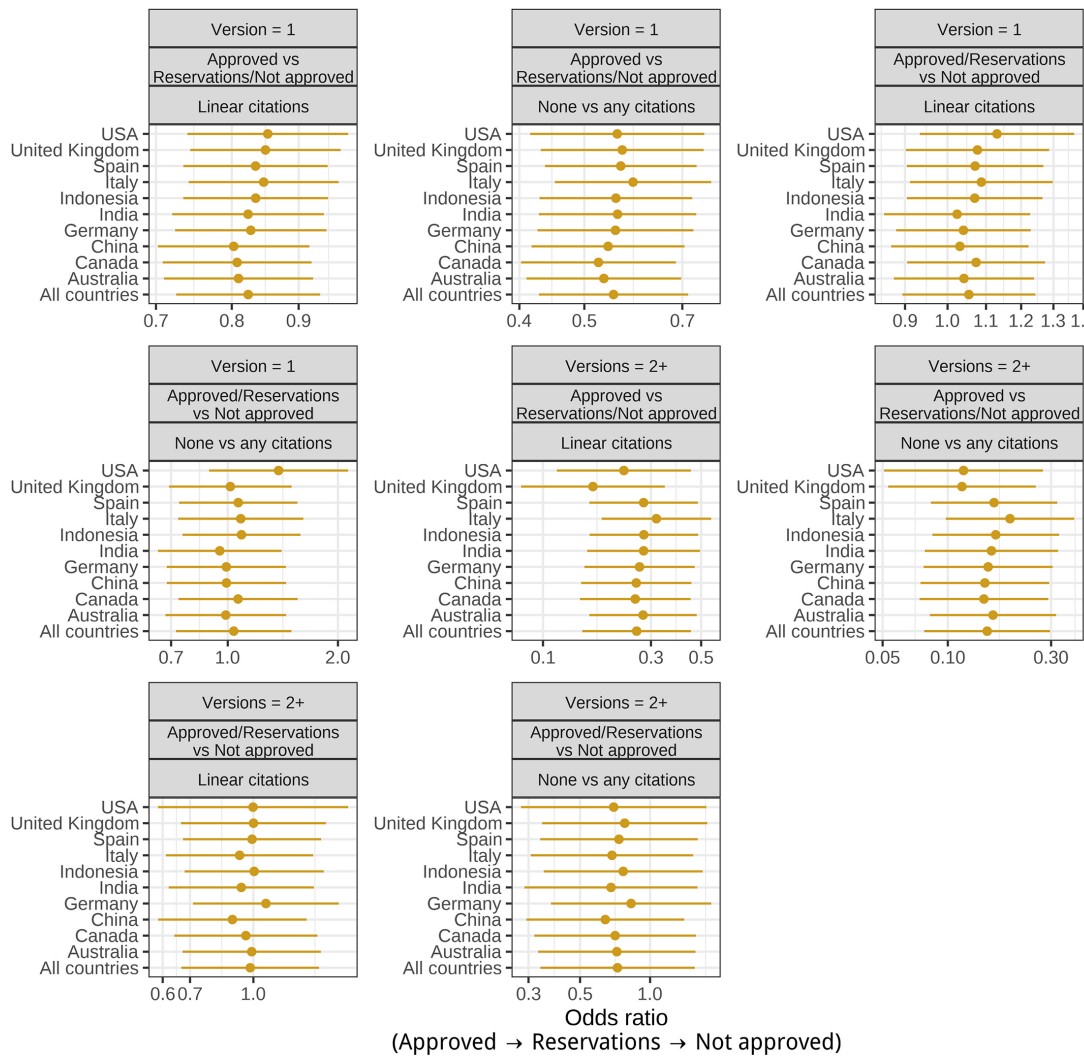

**Appendix 6—figure 4.** Leave-one-country-out sensitivity analyses for research question 2. Odds ratios and adjusted 99.4% confidence intervals for whether the reviewer gave a more or less favourable recommendation when they included a citation to their own articles. The results are shown for the combinations of predictor variables (linear or any vs none), outcome (Approved → Reservations → Not approved) and article version.

# Appendix 7

## Examples of reviewers' requests to cite their own articles

**Appendix 7—table 1.** Example sentences that reviewers used when suggesting citations to their own articles using a random sample of 20 reviews.
The first column shows the number of citations suggested. We have removed any references to names using [xxxx]. The results are ordered by text length.

| Citations suggested | Reviewer's text |
| --- | --- |
| 3 | Also, the introduction, main discussion, and conclusion must be redrawn to highlight NO as a treatment option, the clinical trials discussion, the use of several NORMS (NORM-1, NORM-2, etc.), the effect of NO-carriage system, Natural NO-sources, synthetic NO-sources with limitations, Inorganic versus organic forms, etc. (e.g., in the review publications as given). |
| 1 | The term 'true bugs' applies to the monophyletic Heteroptera, which does include the species presented here (Acanthosoma haemorrhoidale), while aphids and mealybugs belong to the distinct lineage Sternorrhyncha, sometimes (formerly) regarded as a part of the paraphyletic Homoptera (see, for example, *Figure 2* in: [xxxx]). |
| 1 | On a side note: There is already published work on population genomics of the European plaice showing that two large chromosomal rearrangements (two putative inversions) segregate in northern plaice populations (North Sea, Baltic Sea, Barents Sea, and Iceland) and distinguish different plaice populations. |
| 1 | There is some observational clinical data on how the detrusor compensates for the growing prostate and the-by consequence-increase in bladder outflow obstruction, in addition to the animal studies referred to in the commentary, to explain the pathophysiology. |
| 1 | I would like to thank the authors for including references to the work done in the [xxxx] project; I would recommend to remove the reference ([xxxx] et al., 2019b) and replace it with a reference to a much more recent and related article ([xxxx] et al.): |
| 2 | The cited literature is incomplete; it does not include all reports of studies on the presence of the snail in Colombia, and studies with relevant findings of nematodes with or without pathogenic potential in animals are omitted e.g.: |
| 1 | In the last 2 decades, our group has developed a brief instrument to assess the presence and the severity of sensory phenomena (the University of [xxxx]) to investigate OCD phenotypic subtypes and its relationship with TS/CTD. |
| 5 | The authors can find the following relevant articles to enhance their Materials and Methods section and incorporate citations to support their revised manuscript. |
| 3 | The authors should consider references from high impact journal publications on crop yield prediction. For example, the following articles by this reviewer |
| 1 | No mention to more rigorous rankings such as the Leiden Ranking are made nor to what exactly rankings are portraying. See for instance [xxxx]. |
| 1 | Maybe 'manifest' and 'not manifest' would work better for example we used this terminology in [xxxx]. |
| 1 | This is especially useful if you have multiple data sets – see, for example, the [xxxx] package. |

*Appendix 7—table 1 Continued on next page*

*Appendix 7—table 1 Continued*

| Citations suggested | Reviewer's text |
|---|---|
| 3 | I would suggest the authors include some of the results of a large-scale project in Europe. |
| 2 | Please refer to some further references to revise the relevant description: |
| 1 | Here are a few additional publications you might consider referencing. |
| 1 | However, genomics resources are limited, except for parasitoid wasps. |
| 1 | Refer to this recent literature review. |
| 4 | *No relevant sentence* |
| 1 | *No relevant sentence* |
| 6 | *No relevant sentence* |

## Appendix 8

### Views of reviews

We randomly sampled 200 reviews from our sample and collected the number of times the review had been viewed online. A histogram of view counts is shown in *Appendix 8—figure 1*, which had a strong positive skew with most reviews having 10 or fewer views. We used a Poisson model to estimate the annual number of views per year, accounting for the reviews' publication dates. The mean number of views per year was 1.24 with a 95% credible interval of 1.20–1.28.

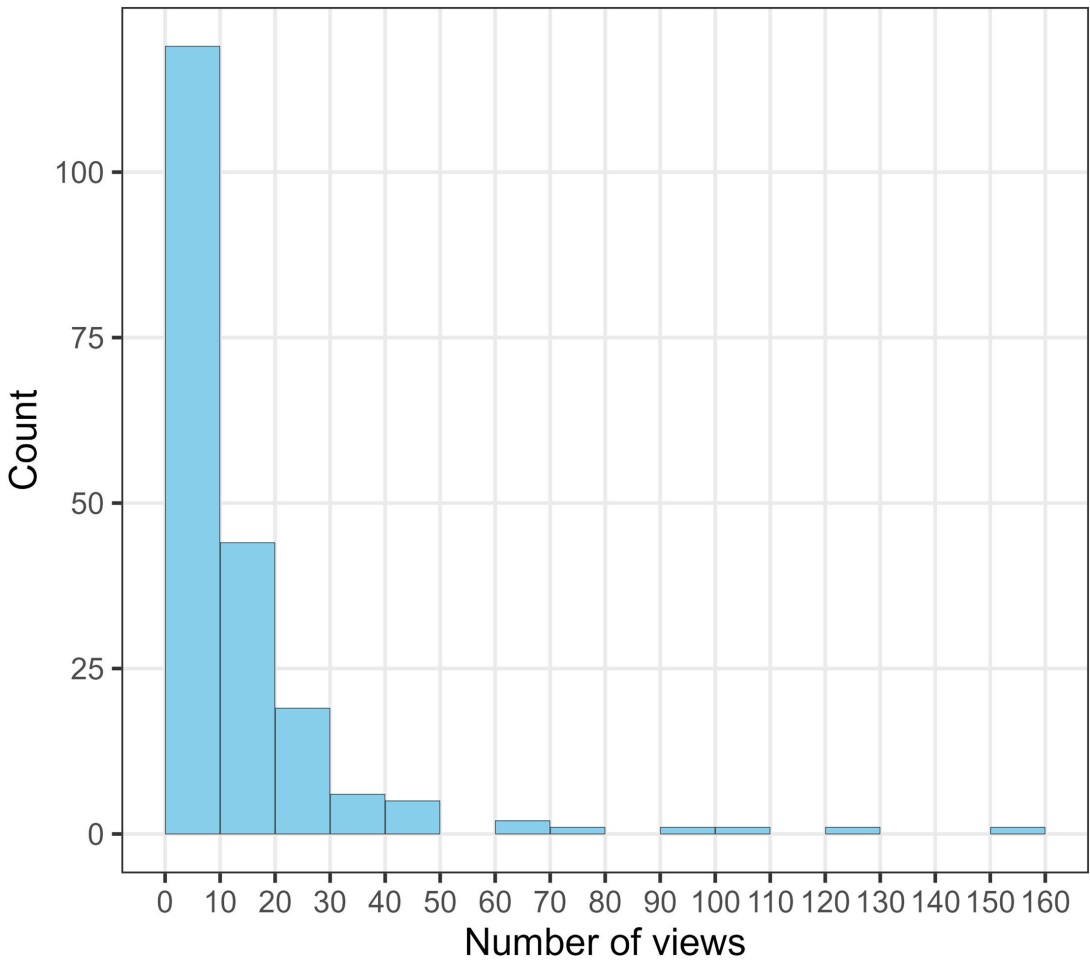

**Appendix 8—figure 1.** Histogram of online view counts of published reviews. The bins are in tens starting at [0, 10).

## Appendix 9

### Data validation

We randomly selected reviews from our analysis data and manually verified the accuracy of our automated data extraction. We checked the accuracy of:

- Reviewers that were cited
- Reviewers that were not cited
- Reviewers that included citation(s) to their own articles in their review

We used a Bayesian calculation to estimate the error rates of our data extraction. We started with a vaguely informative Beta(1, 3.32) prior, which had a 90% probability that the error rate was under 0.5. This vague prior was used to exclude high error rates which were unlikely given our testing of the code during the construction of the data extraction. We created posterior estimates for the error rates using the observed counts of errors from manual checks. We calculated the 90% limits for the posterior distributions as an upper estimate of the error rates.

The distributions are plotted in *Appendix 9—figure 1* and the error rates are shown in *Appendix 9—table 1*. The errors are proportions, with 0 for no errors and 1 for all errors. The highest error rate was for citation(s) to their own articles.

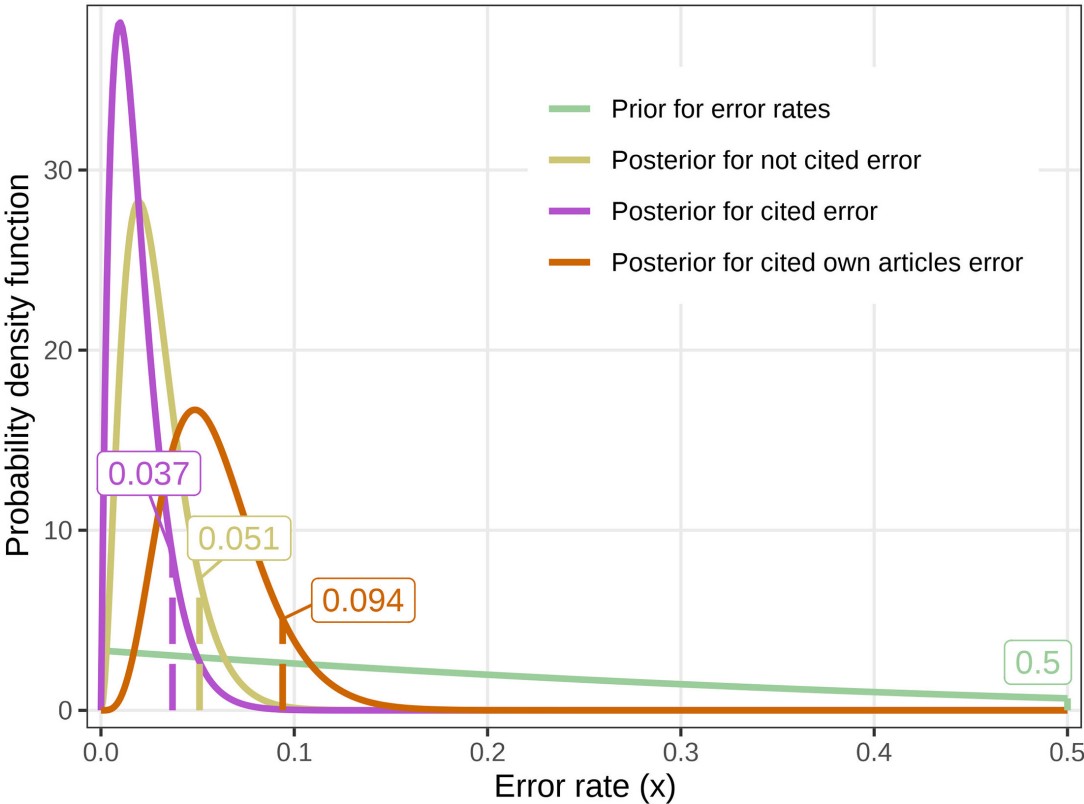

**Appendix 9—figure 1.** Distributions of the error rates. Vaguely informative prior and posteriors for errors for not cited reviewers, cited reviewers, and self-citations. The dashed vertical lines are at Pr(error $\leq x$) = 90%.

**Appendix 9—table 1.** Number of errors found in our data extraction algorithm from manual checks and the estimated 90% limit for the error rate.

| Check | Number checked | Errors found | Pr(Error rate $\leq x$) = 90% |
|---|---|---|---|
| Reviewer not cited | 100 | 2 | 0.051 |
| Reviewer cited | 100 | 1 | 0.037 |
| Reviewer's citation to their own articles | 80 | 4 | 0.094 |

The two errors for reviewers not being cited were for citations to a book and a conference paper that did not have a DOI. All four errors in capturing self-citations were where the number captured was fewer than the true number; for example, we extracted 1 self-citation when the true number was 3.

