## [Editor Report · eLife Assessment]

This **important** study explored a number of issues related to citations in the peer review process. An analysis of more than 37000 peer reviews at four journals found that: (i) during the first round of review, reviewers were less likely to recommend acceptance if the article under review cited the reviewer's own articles; (ii) during the second and subsequent rounds of review, reviewers were more likely to recommend acceptance if the article cited the reviewer's own articles; (iii) during all rounds of review, reviewers who asked authors to cite the reviewer's own articles (a practice known as 'coercive citation') were less likely to recommend acceptance. However, when an author agreed to cite work by the reviewer, the reviewer was more likely to recommend acceptance of the revised article. The evidence to support these claims is **convincing**.

---

## [Referee Report · Joint Public Review]

From Reviewer 3 previously: Barnett examines a pressing question regarding citing behavior of authors during the peer review process. In particular, the author studies the interaction between reviewers and authors, focusing on the odds of acceptance, and how this may be affected by whether or not the authors cited the reviewers' prior work, whether the reviewer requested such citations be added, and whether the authors complied/how that affected the reviewer decision-making.

Key findings are (a) that reviewers were more likely to approve an article if cited in the submission, (b) reviewers who requested a citation in an updated version were less likely to approve, and (c) reviewers who requested and received a citation were more likely to approve the revised version.

Comment from the Reviewing Editor about the latest version:

This is the third version of this article. Comments made during the peer review of the second version, along with author's responses to these comments, are available below.

Comments made during the peer review of the first version, along with author's responses to these comments, are available with previous versions of the article.

---

## [Author Response]

The following is the authors’ response to the previous reviews.

**Editors comments:**
I would encourage you to submit a revised version that addresses the following two points:[a] The point from Reviewer #1 about a possible major confounding factor. The following article might be germane here: Baas and Fennell, 2019: https://papers.ssrn.com/sol3/papers.cfm?abstract_id=3339568

I don’t believe that the point raised by reviewer 1 is a confounder, see my response below.

This article highlighted was in my reading list, but I did not cite it because I was confused by its methods.

The point from Reviewer #4 about the abstract. It is important that the abstract says something about how reviewers reacted to the original versions of articles in which they were cited (ie, the odds ratio = 0.84, etc result), before going on to discuss how they reacted to revised articles (ie, the odds ratio = 1.61, etc result). I would suggest doing this along the following lines - but please feel free to reword the passage "but this effect was not strong/conclusive":When reviewers were cited in the original version of the article under review, they were less likely to approve the article compared with reviewers who were not cited, but this effect was not strong/conclusive (odds ratio = 0.84; adjusted 99.4% CI: 0.69-1.03). However, when reviewers were cited in the revised version of the article, they were more likely to approve compared with reviewers who were not cited (odds ratio = 1.61; adjusted 99.4% CI: 1.16-2.23).

I have changed the abstract to include the odds ratios for version 1 and have used the same wording as from the main text.

**Reviewer #1 (Public review):**
Summary:The work used open peer reviews and followed them through a succession of reviews and author revisions. It assessed whether a reviewer had requested the author include additional citations and references to the reviewers' work. It then assessed whether the author had followed these suggestions and what the probability of acceptance was based on the authors decision. Reviewers who were cited were more likely to recommend the article for publication when compared with reviewers that were not cited. Reviewers who requested and received a citation were much likely to accept than reviewers that requested and did not receive a citation.Strengths and weaknesses:The work's strengths are the in-depth and thorough statistical analysis it contains and the very large dataset it uses. The methods are robust and reported in detail.I am still concerned that there is a major confounding factor: if you ignore the reviewers requests for citations are you more likely to have ignored all their other suggestions too? This has now been mentioned briefly and slightly circuitously in the limitations section. I would still like this (I think) major limitation to be given more consideration and discussion, although I am happy that it cannot be addressed directly in the analysis.

This is likely to happen, but I do not think it’s a confounder. A confounder needs to be associated with both the outcome and the exposure of interest. If we consider forthright authors who are more likely to rebuff all suggestions, then they would receive just as many citation and self-citation requests as authors who were more compliant. The behaviour of forthright authors would likely only reduce the association seen in most authors which would be reflected in the odds ratios.

**Reviewer #2 (Public review):**
Summary:This article examines reviewer coercion in the form of requesting citations to the reviewer's own work as a possible trade for acceptance and shows that, under certain conditions, this happens.Strengths:The methods are well done and the results support the conclusions that some reviewers "request" self-citations and may be making acceptance decisions based on whether an author fulfills that request.Weakness:I thank the author for addressing my comments about the original version.
**Reviewer #3 (Public review):**
Summary:In this article, Barnett examines a pressing question regarding citing behavior of authors during the peer review process. In particular, the author studies the interaction between reviewers and authors, focusing on the odds of acceptance, and how this may be affected by whether or not the authors cited the reviewers' prior work, whether the reviewer requested such citations be added, and whether the authors complied/how that affected the reviewer decision-making.Strengths:The author uses a clever analytical design, examining four journals that use the same open peer review system, in which the identities of the authors and reviewers are both available and linkable to structured data. Categorical information about the approval is also available as structured data. This design allows a large scale investigation of this question.Weaknesses:My original concerns have been largely addressed. Much more detail is provided about the number of documents under consideration for each analysis, which clarifies a great deal.Much of the observed reviewer behavior disappears or has much lower effect sizes depending on whether "Accept with Reservations" is considered an Accept or a Reject. This is acknowledged in the results text. Language has been toned down in the revised version.The conditional analysis on the 441 reviews (lines 224-228) does support the revised interpretation as presented.No additional concerns are noted.
**Reviewer #4 (Public review):**
Summary:This work investigates whether a citation to a referee made by a paper is associated with a more positive evaluation by that referee for that paper. It provides evidence supporting this hypothesis. The work also investigates the role of self-citations by referees where the referee would ask authors to cite the referee's paper.Strengths:This is an important problem: referees for scientific papers must provide their impartial opinions rooted in core scientific principles. Any undue influence due to the role of citations breaks this requirement. This work studies the possible presence and extent of this.The methods are solid and well done. The work uses a matched pair design which controls for article-level confounding and further investigates robustness to other potential confounds.Weaknesses:The authors have addressed most concerns in the initial review. The only remaining concern is the asymmetric reporting and highlighting of version 1 (null result) versus version 2 (rejecting null). For example the abstract says "We find that reviewers who were cited in the article under review were more likely to recommend approval, but only after the first version (odds ratio = 1.61; adjusted 99.4% CI: 1.16 to 2.23)" instead of a symmetric sentence "We find ... in version 1 and ... in version 2".

The latest version now includes the results for both versions.